# Advanced Technologies for Nitrogen Removal and Recovery from Municipal and Industrial Wastewater

**DOI:** 10.3390/ma18071422

**Published:** 2025-03-23

**Authors:** Sławomir Kasiński, Przemysław Kowal, Krzysztof Czerwionka

**Affiliations:** 1Faculty of Geoengineering, University of Warmia and Mazury in Olsztyn, Prawocheńskiego Street 15, 10-720 Olsztyn, Poland; slawomir.kasinski@uwm.edu.pl; 2Faculty of Civil and Environmental Engineering, Gdansk University of Technology, Narutowicza Street 11/12, 80-233 Gdansk, Poland; kczer@pg.edu.pl

**Keywords:** nitrogen removal, nitrification–denitrification, anammox process, shortcut nitrogen pathways, ammonia stripping, membrane technologies, nitrogen recovery, environmental sustainability, advanced wastewater treatment

## Abstract

Nitrogen pollution poses significant environmental challenges, contributing to eutrophication, soil acidification, and greenhouse gas emissions. This study explores advanced methods for nitrogen removal and recovery from municipal and industrial wastewater, with a focus on biological, chemical, and physical processes. Key processes, such as nitrification–denitrification and emerging technologies like shortcut nitrogen pathways, were analyzed for their efficiency, cost-effectiveness, and environmental benefits. This review highlights the integration of innovative techniques, including membrane systems and ammonia stripping, with traditional approaches to enhance nitrogen management. Emphasis is placed on optimizing operational conditions, such as pH, temperature, and carbon-to-nitrogen ratios, to achieve high removal rates while minimizing energy consumption and environmental impact. These findings underline the critical role of interdisciplinary strategies in addressing the challenges of nitrogen pollution and promoting sustainable wastewater management.

## 1. Introduction

### 1.1. Origin and Consequences of Nitrogen Emissions in Municipal and Industrial Wastewater

Anthropogenic nitrogen pollution of the environment results from industrialized processes that fix atmospheric nitrogen (N_2_) which, in its original form, is biologically unavailable to most organisms. These processes convert nitrogen into reactive forms, which are crucial for food production [1], but simultaneously contribute to water eutrophication, soil acidification, and nitrous oxide emissions, the latter being a greenhouse gas. Estimates from 1995 indicated that biotic nitrogen fixation on land ranged from 90 to 130 Tg per year (1 Tg = 10^12^ g), while anthropogenic activities added an additional 140 Tg of nitrogen annually, with 20 Tg from fossil fuel combustion, 80 Tg from fertilizer production, and 40 Tg from the cultivation of nitrogen-fixing crops, such as legumes and rice [2]. Current estimates show that anthropogenic production of reactive nitrogen has increased to approximately 210 Tg per year, about twice the natural nitrogen fixation in terrestrial ecosystems [3]. Considering that nitrogen was historically recovered from natural sources, such as animal manure and wastewater, the widespread use of synthetic nitrogen fertilizers has radically changed nitrogen management. Projections for the year 2100 suggest a further increase in anthropogenic reactive nitrogen production. Depending on scenarios, including the rate of human population growth, nitrogen production in the fertilizer sector alone could rise to as much as 172 Tg per year [4]. These figures underscore the exponential increase in human impact on the nitrogen cycle and the urgent need to implement measures to reduce reactive nitrogen emissions and their negative consequences. This is particularly important since only about half of anthropogenic nitrogen can be accurately accounted for, while the remainder stays on continents in groundwater, soils, or vegetation, or is denitrified back to N_2_ [5].

The forms of nitrogen found in municipal and industrial wastewater can be broadly classified as inorganic and organic. The primary inorganic nitrogen forms in wastewater are ammonia and its ionized form, ammonium ion (NH_3_/NH_4_^+^), nitrites (NO_2_^−^), and nitrates (NO_3_^−^). According to the requirements of Directive 91/271/EEC concerning urban wastewater treatment, the total nitrogen concentration in treated wastewater (i.e., the sum of ammonia, nitrates, and nitrites) must not exceed 10 mg/L for wastewater treatment plants in the EU with a capacity exceeding 10,000 PE (population equivalent) and 15 mg/L for treatment plants with a capacity ranging from 2000 to 10,000 PE. Typical concentrations of the nitrogen species at the municipal level are summarized in Table 1.

The synthesis of ammonia in natural conditions occurs as a result of a series of biochemical processes related to nitrogen metabolism. Significant processes of ammonia synthesis in living organisms include the deamination of amino acids during the catabolism of proteins and amino acids [7], and ammonification, carried out by saprophytic bacteria and fungi during the decomposition of dead organic matter [8]. As a result, the main sources of ammonia in municipal wastewater are human metabolic products. Urine and feces contain substantial amounts of urea, creatinine, and other nitrogen compounds, which undergo hydrolysis and deamination to form ammonia [9]. An additional source of ammonia in wastewater is amino acids present in flushed food waste, such as leftover food. Cleaning agents and detergents are also significant, as some chemical products contain nitrogen compounds that can release ammonia upon degradation. The concentration of ammonia in municipal wastewater typically ranges from 20 to 75 mg/L NH_4_^+^-N [10].

The industrial synthesis of ammonia primarily occurs via the Haber–Bosch process, in which atmospheric nitrogen (N_2_) reacts with hydrogen (H_2_) under high pressure and temperature in the presence of a catalyst to form ammonia. The discovery of this process was a breakthrough in chemistry and chemical engineering, enabling the production of nitrogen fertilizers on the one hand and explosives on the other [11]. Ammonia concentrations in fertilizer plants at various production points can range from several hundred to as high as 2500 mg/L NH_4_^+^-N [12]. Industrially produced ammonia is also used in the textile industry (17–273 mg/L NH_4_^+^-N) [13] and the pharmaceutical industry (40–320 mg/L NH_4_^+^-N) [14].

The food industry, particularly meat and fish processing and dairy production, is another source of ammonia. According to [14], typical ammonia concentrations in wastewater from meat processing range from 20 to 300 mg/L NH_4_^+^-N, with similar levels observed in the dairy industry, and lower concentrations in the fruit and vegetable processing sector. In the pulp and paper industry, associated with paper production, ammonia concentrations range from a few to several dozen mg/L NH_4_^+^-N, depending on the level of wastewater dilution [15].

Under natural conditions, nitrites (NO_2_^−^) and nitrates (NO_3_^−^) are formed through a two-step nitrification process carried out by autotrophic nitrifying bacteria, such as those of the genera Nitrosomonas and Nitrobacter. On a global scale—despite the fact that the worldwide production of municipal wastewater is approximately 360 billion m^3^ annually [16]—most nitrates are generated in soil environments. This is primarily due to the vast area covered by agricultural fields and natural soil ecosystems. Typical nitrate concentrations in municipal wastewater following the nitrification process were estimated by [10] at around 28 mg N/L, although these values may vary depending on system load and operational conditions. Nitrite concentrations during nitrification are usually low due to their relatively rapid conversion to nitrates. However, in most biological wastewater treatment systems, nitrate emissions are significantly limited by the denitrification process.

The synthesis of most nitrates in industrial conditions (e.g., metal nitrates or ammonium nitrate) is primarily based on the production of nitric acid (i.e., through the Ostwald process), which then reacts with target compounds such as ammonia, hydroxides, and carbonates. Globally, nitrate synthesis is mainly associated with the production of nitrate fertilizers. The type of nitrate compound used depends on the crop type, soil type, and local agronomic and environmental requirements. One of the most common nitrate fertilizers is ammonium nitrate (NH_4_NO_3_), which provides nitrogen in two forms: ammonium (NH_4_^+^) and nitrate (NO_3_^−^). This duality allows ammonium nitrate to provide both rapid and sustained nitrogen release, making it a versatile fertilizer for various crops.

In the food industry, nitrates and nitrites are used as food preservatives, especially in meat processing (e.g., curing salts), fish, and certain dairy products. According to the Food and Feed Information Portal Database [17], these preservatives include four compounds: potassium nitrite (E249), sodium nitrite (E250), sodium nitrate (E251), and potassium nitrate (E252). Nitrates are also widely used in the textile, dye, and pharmaceutical industries, as well as in the synthesis of materials, such as rubbers. The emission of nitrates and nitrites into industrial wastewater typically occurs at various production stages, but areas of emissions can also include loading and storage sites prone to leaching by rainwater [18]. The concentrations of nitrates and nitrites in wastewater can be significant, posing challenges for local wastewater treatment plants. For example, an industrial facility in Sweden, as described by [19], intensively used calcium nitrate tetrahydrate (Ca(NO_3_)_2_·4H_2_O) as a coagulant in the production of medical items such as anesthesia bags and catheter balloons (consumption of approximately 28 tons per year). The average nitrate concentration in wastewater was 140 mg/L, and the average nitrite concentration was around 3 mg/L. Interestingly, the inhibitory effect of nitrites on nitrification in the treatment plant was proportionally greater than that of nitrates.

Organic nitrogen (ON) in wastewater occurs in the form of compounds with varying sizes, solubility, and levels of chemical complexity. It is important to note that different authors use diverse criteria for defining size fractions in wastewater with respect to their solubility. For example, ref. [20] defined soluble particles as <0.1 µm, colloidal particles as 0.1–1 µm, supra-colloidal particles as 1–12 µm, and settleable particles as >12 µm. In contrast, ref. [21] adopted boundaries of <0.22 µm for soluble fractions and >0.22 µm for colloidal and supra-colloidal fractions. Some authors also use atomic mass units (amu; 1 amu = 1 Da = 1.66·10^−24^ g) for classification. For instance, ref. [22] defined soluble particles as <10,000 amu, colloidal particles as ranging from 10,000 amu to 1 µm, and supra-colloidal particles as >1 µm. One amu corresponds to 1/12 of the mass of a carbon-12 atom, approximately 1.66·10^−24^ g.

Among dissolved organic nitrogen (DON) forms, ref. [23] identified free amino acids—derived from protein degradation and organism metabolism; bound amino acids—present in proteins, peptides, and adsorbed onto humus; urea—a product of bacterial and animal metabolism, as well as nucleotides and nucleic acids (DNA and RNA)—originating from dead organisms. The authors also highlight the presence of sugar amines, such as glucosamine, resulting from the degradation of chitin and bacterial and fungal cell walls, as well as methylamines, which are characteristic of anaerobic environments. Humic substances, although considered less biologically active, also deserve attention as they can provide nitrogen to microorganisms. Dissolved organic nitrogen forms thus represent a diverse group of compounds with critical importance in the nitrogen cycle. Studies indicate that a significant majority of DON in municipal wastewater (approximately 80%) is hydrophilic, which poses challenges for its removal through physical processes such as adsorption [24].

The colloidal and suspended organic nitrogen fraction includes larger organic particles containing nitrogen, such as fragments of microbial cells, food residues, humic materials, and other colloidal substances. Research indicates that colloidal organic nitrogen can constitute up to 68% of the total organic nitrogen in wastewater streams [25]. The group of organic nitrogen compounds with sedimentation properties includes larger fragments of organic matter, such as plant fibers, food residues, fecal particles, and other solid fractions of biological and anthropogenic origin.

There is also a group of nitrogen-containing compounds that are resistant to biological degradation. These include pesticides, pharmaceuticals, hormones, and other nitrogen-containing xenobiotics [26,27,28]. One of the most comprehensive studies in this area is the work of [29], which analyzed the presence of 156 polar organic compounds in samples from 90 municipal wastewater treatment plants across Europe. The authors examined a wide range of substances, including pharmaceuticals (e.g., carbamazepine, tramadol, diclofenac), plant protection products, personal care products, flame retardants, medical imaging contrast agents, and perfluoroalkyl substances. The study revealed that most of these compounds were present in both influent and effluent from wastewater treatment plants, with influent concentrations often ranging from nanograms to micrograms per liter. These findings are broadly consistent with reports from other parts of the world [28,30,31]. A characteristic feature observed was the diurnal variation in concentrations of hormones (estrone, estriol) and antibiotics (trimethoprim, sulfamethoxazole) in influent, with peak concentrations in the morning hours, reflecting human metabolic activity [32]. Although the concentrations of these compounds in wastewater are not high, their presence can lead to endocrine disruptions in aquatic ecosystems [33].

Certain industries, such as coke production, metallurgy, electroplating, electrochemical processes, petrochemicals, and pharmaceuticals, can also be sources of cyanides (CN^−^), aromatic nitro compounds such as nitrobenzene and nitrotoluene, and nitrogen-containing heterocyclic compounds such as pyridine, quinoline, and indole. The concentration of these compounds in wastewater emissions is closely linked to the type of industrial activity. Cyanide primarily exists in its free form (CN^−^) at pH levels above 8.7 or as derivative compounds, such as cyanates, nitriles, or metal–cyanide complexes. Under acidic conditions, cyanide can form hydrogen cyanide. Thus, the characteristics of the industrial process itself can influence the nature of its emissions. For example, studies on wastewater from a coke plant [34] reported high total nitrogen concentrations (~240 mg/L), including 7.5% cyanide and 40.4% thiocyanate. Additionally, 76 types of nitrogen-containing heterocyclic compounds were detected. In wastewater from the metallurgical industry [35], cyanide concentrations were reported at 4.84 mg/L and <0.2 mg/L after treatment. It is noteworthy that the jewelry industry in Córdoba, Spain produces 4–5 tons of alkaline waste annually, containing up to 26 g/L of free cyanide (~1 M) along with significant amounts of heavy metals [36]. Obvious sources of nitro compounds include facilities involved in manufacturing pyrotechnics and explosives. Studies of wastewater from munitions factories have revealed the presence of polar metabolites of nitro compounds, such as nitrobenzoates and aminobenzoates, at concentrations reaching several hundred micrograms per liter [37]. Similar concentrations were described by [38] in condensate water generated during the production of 2,4,6-trinitrotoluene (TNT). Nitrogen-containing heterocyclic compounds are produced in various sectors of chemical, agrochemical, and pharmaceutical manufacturing. However, common sources include wastewater from the petroleum and coal refining industries [39]. In studies by [40], concentrations of compounds such as carbazole, fluorene, and dibenzothiophene in refinery wastewater reached up to 10 mg/L for each compound.

Nitrogen can form complexes with heavy metals such as copper, nickel, and zinc, which are present in industrial wastewater. The mechanisms for synthesizing these compounds vary but result in chemically stable complexes that are resistant to conventional treatment methods. For example, copper–ammonia complexes are stabilized by hydrogen bonding and electrostatic attraction between copper ions and ammonia molecules [41]. Similarly, cyanide–metal complexes, such as ferricyanide [Fe(CN)_6_]^3−^ and tetracyanonickelate [Ni(CN)_4_]^2−^, exhibit comparable chemical stability [42].

The final product of denitrification processes is molecular nitrogen (N_2_), where nitrates and nitrites are reduced to gaseous nitrogen by denitrifying bacteria under anaerobic conditions. This process is critical for the biological removal of nitrogen from wastewater, allowing its safe release into the atmosphere. This stage can be considered a symbolic closure of the nitrogen cycle.

### 1.2. Methods of Nitrogen Input into the Environment

Methods of nitrogen input into the environment can generally be divided into point sources and diffuse sources. In the case of point source emissions, the release of nitrogen compounds into the environment occurs when treatment processes are not fully effective (assuming they are implemented at all). If wastewater treatment methods are either ineffective or improperly balanced, environmental damage is likely, even with significant technological and infrastructural investments. An example of this can be seen in Eastern China, where, between 2008 and 2017, approximately 35 billion m^3^ of municipal wastewater treatment capacity was constructed. The implemented technologies were highly effective in removing phosphorus (about 90%) but had limited efficiency in nitrogen removal (60–70%). As a result, the TN/TP ratio in treated wastewater increased from a median of 10.7 to 17.7, leading to changes in aquatic ecosystem composition. This shift promoted the proliferation of toxic cyanobacterial species, such as *Microcystis* spp. and *Planktothrix* spp. [43].

An aggravation of point source nitrogen emissions into the environment comes from illegal or uncontrolled wastewater discharges. These practices, often bypassing any form of treatment, introduce large quantities of nitrogen and phosphorus compounds into water bodies, contributing to the eutrophication of aquatic ecosystems. This issue is particularly significant in regions with high population density and intensive water resource usage. In areas such as the Mediterranean Sea, direct discharges of domestic wastewater constitute a major source of nutrients for coastal waters. In 2003, such discharges contributed approximately 15·10^9^ mol N per year, comparable to nutrient loads delivered by rivers. Projections indicate that, without remedial action, these emissions could increase by 254% by 2050 in the southern Mediterranean coastal countries [44]. Illegal discharges also contribute to biological toxicity, as demonstrated in Canada. There, emissions of reactive nitrogen forms, such as nitrates and ammonia, increased between 2003 and 2008, despite upgrades to treatment systems. In some regions, toxicity was further exacerbated by the presence of heavy metals and other chemical compounds [45].

When discussing diffuse nitrogen emissions into the environment, it is essential to highlight the impact of agricultural activities, including livestock farming. Excessive application of nitrogen fertilizers, both mineral and organic, leads to environmental pollution through surface runoff from agricultural fields. The dominant form of nitrogen in runoff from agricultural fields is nitrate nitrogen [46]. Nitrates (NO_3_^−^) are among the most widespread forms of nitrogen in surface and groundwater due to their high solubility in water and weak binding to soil particles. As negatively charged ions, nitrates are not effectively retained by soil substrates and can be leached with rainwater, eventually reaching groundwater. Consequently, they represent a significant nitrogen source for aquatic ecosystems, especially in regions with intensive agriculture where nitrogen fertilizers are overapplied. Organic nitrogen and ammonium nitrogen are typically present in smaller proportions but can be significant in areas where organic fertilizers are heavily used. Livestock farming is a major source of ammonia emissions to the environment. For instance, in Europe, livestock production accounts for an estimated 75% of total ammonia emissions [47]. Ammonia emissions originate from various stages of livestock production, including animal housing, manure storage, manure application to fields, and excretions from grazing animals. The nitrogen fertilizer industry, alongside agriculture and livestock farming, is one of the most significant sources of diffuse nitrogen emissions. Monitoring studies conducted near fertilizer factories have clearly shown substantial increases in nitrate concentrations in soil, drainage water, and surface water. For example, around a factory in Talkha, Egypt, nitrate concentrations were highest in close proximity to the pollution source and decreased significantly with distance [48].

At the same time, it is worth noting that nitrogen concentrations in surface waters and soil undergo significant seasonal fluctuations. In spring, as temperatures begin to rise and precipitation increases, surface runoff intensifies, leading to higher nitrogen concentrations in surface waters. In summer, however, due to higher temperatures and increased microbiological and biochemical activity in the soil, the emission of nitrogen oxides (NO_x_) into the atmosphere rises. Seasonal changes in the environment can be closely linked to variations in the hydrological cycle, as confirmed by research conducted on various aquatic and terrestrial ecosystems. O’Connell used isotope analysis and hydrological modeling to determine the impact of anthropogenic activity on the nitrogen cycle, demonstrating that agricultural intensification and land-use changes significantly increase nitrate concentrations in surface waters, particularly during periods of increased precipitation and low river flows, when nitrogen accumulates and is transported more slowly through river systems [49]. Changes in the water balance also affect the emission of gaseous forms of nitrogen from wetland ecosystems. Liu et al. demonstrated that fluctuations in water levels in wetlands influence vegetation distribution and microbiological activity in sediments, regulating the emission of nitrous oxide (N_2_O), a potent greenhouse gas. Studies conducted in Dongting Lake showed that flood conditions promote denitrification, whereas periodic drying of sediments leads to nitrogen accumulation and increased N_2_O emissions into the atmosphere [50]. In China, reactive nitrogen emissions from the soil, including nitrogen oxides and nitric acid, contribute to increased ozone levels during the summer, which is the result of interactions between soil processes and reduced anthropogenic emissions. Paradoxically, summer NO_x_ emissions from the soil into the atmosphere are primarily driven by changes in the hydrological regime and soil oxygen availability, which influence the intensity of nitrification and denitrification processes [51].

In response to growing concerns about the impact of changes in the nitrogen cycle on the environment, environmental policy has increasingly focused on implementing measures to reduce reactive nitrogen emissions and improve its efficiency in agriculture, industry, and water management. The excessive release of nitrogen into ecosystems contributes to soil degradation, water eutrophication, and increased greenhouse gas emissions, making the development of comprehensive nitrogen management strategies essential. Research indicates that nitrogen management regulations should take into account economic and social factors, as well as changing climatic conditions and the growing global population. In light of these findings, Kanter et al. proposed new policy models based on nitrogen emission trajectories within the framework of Shared Socioeconomic Pathways (SSPs), considering the impact of climate change, population growth, and urbanization on future nitrogen management needs [52]. Their analysis indicates that the effectiveness of nitrogen policy will strongly depend on the level of regulatory ambition and global coordination of actions. In the SSP1 (sustainable development) scenario, significant reductions in nitrogen emissions can be achieved through more efficient fertilization, nitrogen waste recycling, and reduced consumption of animal protein. In contrast, in less optimistic pathways, such as SSP3 (regional rivalry) and SSP5 (intensive industrial development), continued nitrogen resource exploitation without proper regulation may lead to catastrophic environmental consequences. The results of these studies highlight the urgent need to implement a coordinated global nitrogen policy, integrating agricultural, industrial, and climate strategies to mitigate the negative impact of excessive nitrogen emissions on the environment. Appropriate regulations should include policies aimed at reducing nitrous oxide (N_2_O) emissions—a potent greenhouse gas—as well as measures to prevent nitrate contamination of groundwater and surface water. International cooperation mechanisms for monitoring nitrogen emissions and promoting technologies that reduce nitrogen losses in ecosystems are also of key importance. Only through a comprehensive and long-term approach will it be possible to effectively mitigate the negative effects of anthropogenic disruptions in the nitrogen cycle and their impact on human health and ecosystem stability.

### 1.3. Environmental Impact and Toxicity of Various Nitrogen Forms

In the context of the previously discussed sources and forms of nitrogen emissions into the environment, it is important to consider the environmental effects caused by various nitrogen forms. This study focuses primarily on ammonia, nitrites, and nitrates, as they constitute the majority of anthropogenic nitrogen emissions. Each of these nitrogen forms is characterized by different toxicity levels, environmental mobility, and modes of impact on aquatic biocenoses and humans.

Ammonia is considered one of the more toxic forms of nitrogen. In aquatic environments, ammoniacal nitrogen exists primarily in two forms: ionized (NH_4_^+^) and unionized (NH_3_). The sum of these nitrogen forms is referred to as total ammonia nitrogen (TAN). The equilibrium reaction between ammonia forms is represented by the following equation:NH_4_^+^ + OH^−^ ⇄ NH_3_·H_2_O ⇄ NH_3_ + H^+^,(1)

The acid dissociation constant (pK_a_) of the ammonium/ammonia system depends on the pH and temperature of the liquid. For example, at a temperature of 25 °C, pK_a_ is 9.25 (Figure 1).

The toxicity of ammonia primarily results from the presence of its unionized form (NH_3_), which is up to several hundred times more toxic to animals than the ionized form (NH_4_^+^) [53] because it can penetrate cell membranes. This issue is particularly pronounced in aquatic environments. For many fish species, toxic ammonia concentrations begin as low as 0.02–0.05 mg/L [54]. One of the primary toxic effects of ammonia on fish is the disruption of gill epithelial cell function, impairing gas exchange, causing oxidative stress, and leading to further metabolic disturbances. According to [54], excessive ammonia concentrations result in reduced hemoglobin levels and red blood cell counts, ultimately causing tissue damage and physiological dysfunctions. Ammonia bioaccumulation may also induce neurotoxicity through the buildup of glutamine in the brain.

Interestingly, in nutrient-rich aquatic systems—particularly during summer conditions—most cyanobacteria and phytoplankton species prefer ammonia over other nitrogen forms due to its rapid availability and low energy requirements for conversion into usable forms (unlike in animals). Detailed studies demonstrating cyanobacteria’s preference for ammonia were published by Boussiba and Gibson in 1984 [55]. A key metabolic aspect of cyanobacteria appears to be the interdependence of ammonia assimilation with the photosynthesis process. This relationship explains the periodic activity of cyanobacteria in water bodies. However, under favorable environmental conditions (including the availability of nutrients, sunlight, temperature, and mixing), the presence of ammonia is one of the primary factors driving local cyanobacterial blooms [56]. Ongoing eutrophication of water bodies leads to reduced dissolved oxygen levels and pH changes, and over time, it degrades water quality and alters the species composition of the affected ecosystem.

In soil conditions, prolonged exposure of plants to elevated ammonia concentrations can lead to tissue necrosis, reduced growth rates, and disruptions in photosynthesis. According to studies [57], threshold ammonia concentrations in the air, beyond which signs of plant damage occur, are approximately 75 µg/m^3^ (annual average), 600 µg/m^3^ (24 h average), or 10,000 µg/m^3^ (hourly average). As reported by [58], high ammonia concentrations in the air damage plants by disrupting their metabolism and acid-base balance. For example, ammonia accumulation in leaves can acidify cells, impairing enzyme function and membrane processes. This phenomenon is particularly evident in plants growing in acidic or nutrient-poor environments. Reference [58] also noted that ammonia toxicity may result from its ability to disrupt electron transport in chloroplasts, potentially leading to photosynthesis inhibition.

Nitrates (NO_3_^−^) are the final oxidation form of nitrogen compounds in the nitrification process and are one of the most widespread forms of nitrogen in aquatic environments. Unlike nitrites (NO_2_^−^) and ammonia (NH_3_), nitrates have relatively low direct toxicity to aquatic organisms. For instance, reference [59] reported LC50 values for carp (*Cyprinus carpio*) under static conditions of approximately 1075 mg NO_3_^−^-N/L and 0.80 mg NH_3_-N/L. For Siberian sturgeon (*Acipenser baeri*), the 96 h LC50 values for nitrates decreased with fish size, measuring 1028 mg/L for juveniles (6.9 g), 601 mg/L for medium-sized fish (66.9 g), and 397 mg/L for larger fish (673.8 g) [60]. In another study [61], the LC50 value for lake whitefish (*Coregonus clupeaformis*) during 96 h exposure was as high as 1903 mg NO_3_^−^-N/L. The toxic effects of nitrates become more pronounced during long-term exposures and particularly affect juvenile stages of aquatic organisms, where they can cause methemoglobinemia, reducing the blood’s oxygen-carrying capacity. For rainbow trout (*Oncorhynchus mykiss*), chronic nitrate toxicity becomes apparent at concentrations as low as 10 mg NO_3_^−^-N/L [62]. Moreover, as nitrate concentrations in water bodies increase, species diversity declines. According to [63], the critical nitrate threshold for maintaining stable species diversity in water bodies is 0.61–0.64 mg NO_3_^−^-N/L in inflowing water to a lake.

Nitrates, despite being much more soluble in water than ammonia, exhibit relatively low toxicity to aquatic plants. Moreover, the literature emphasizes the role of plants and macrophytes in purifying water bodies of nitrates (e.g., [62]). Common species used for nitrate removal in plant–soil systems include water hyacinth (*Eichhornia crassipes*) [64], water lettuce (*Pistia stratiotes*) [65], elephant grass (*Pennisetum purpureum*) [66], and marsh pennywort (*Hydrocotyle umbellata*) [67]. Additionally, common reed (*Phragmites australis*) demonstrates significant potential in this regard [66,68]. These plants are used not only for wastewater treatment but also for groundwater remediation. The effectiveness of plants in water purification depends on their adaptive capacities. For instance, in studies [69], water hyacinth achieved a high nitrate removal efficiency of 83% at an initial concentration of 300 mg NO_3_^−^-N/L. In another study [70] on the use of Brazilian water milfoil (*Myriophyllum aquaticum*), toxic effects of nitrates were observed only at concentrations of 60 mmol/L (equivalent to 3720.29 mg NO_3_^−^-N/L). According to the authors, the primary toxic effects of high nitrate concentrations on plants include disturbances in osmotic pressure (hindering nutrient assimilation) and, similar to ammonia, oxidative stress, as evidenced by an increase in malondialdehyde (MDA) levels.

Nitrates, due to their high mobility, are particularly prone to leaching deep into the soil profile and eventually reaching groundwater. Research indicates that nitrate loading can affect groundwater quality for decades before nitrates reach the saturated zone and impact local water resources. Independent studies report increasing nitrate concentrations in groundwater across various parts of the world. The average nitrate concentration in England’s groundwater is rising at a rate of 0.34 mg NO_3_^−^-N/L per year [71], while in California, the observed increase is around 0.90 mg NO_3_^−^-N/L per year [72]. In northern Kelantan, Malaysia, 25 years of monitoring revealed an approximate 8.1% increase in nitrate concentrations in shallow aquifers [73]. Many countries have implemented measures to reduce nitrate pollution, such as introducing regulations on fertilizer use and protecting vulnerable zones. In the European Union, the permissible nitrate concentration in drinking water is 50 mg/L, as specified in the Council Directive 98/83/EC on the quality of water intended for human consumption. In the United States, the Environmental Protection Agency (EPA) set the maximum allowable nitrate concentration in drinking water at 10 mg/L. The primary mechanism of nitrate toxicity in humans is related to its reduction to nitrites (NO_2_^−^) and subsequently to nitric oxide (NO). Upon entering the bloodstream, NO oxidizes iron in hemoglobin to the Fe^3+^ state, converting hemoglobin into methemoglobin. This condition is particularly dangerous for newborns and infants, whose methemoglobin reductase system is not fully developed. Consuming water with high nitrate levels can lead to “blue baby syndrome” (methemoglobinemia), characterized by cyanosis, breathing difficulties, and potentially life-threatening effects [74]. Additionally, the literature suggests that nitrates may inhibit iodine uptake by the thyroid and contribute to the formation of carcinogenic N-nitroso compounds [75].

Nitrites (NO_2_^−^) are an intermediate form between ammoniacal nitrogen and nitrate nitrogen in the processes of nitrification and denitrification. They are also one of the most toxic forms of nitrogen in aquatic environments, and their presence at excessively high concentrations can cause numerous adverse effects on aquatic organisms, including developmental and behavioral disturbances. The primary mechanism of nitrite toxicity in fish is their active uptake by chloride cells in the gills, leading to accumulation in blood plasma, often reaching concentrations ten times higher than in the surrounding water. Visual symptoms of nitrite poisoning in fish include the discoloration of gills and blood to brown. As in mammals, the result is the formation of methemoglobin. When methemoglobin levels exceed 70–80% of total hemoglobin, fish become lethargic and may die of hypoxia under stress conditions. Examples of 96 h LC50 values for fish demonstrate a wide range of species sensitivity to nitrites. Species with low nitrite tolerance include rainbow trout (*Salmo gairdneri*), with an LC50 of 0.24 mg/L at low chloride concentrations (0.35 mg Cl^−^/L), and Chinook salmon (*Oncorhynchus tshawytscha*), with an LC50 of 0.7 mg NO_2_^−^-N/L at moderate chloride concentrations. Most fish species show moderate nitrite tolerance, with LC50 values ranging from 5.7 to 22 mg NO_2_^−^-N/L. High-tolerance species include largemouth bass (*Micropterus salmoides*), which has an LC50 of up to 140 mg/L [76]. Methemoglobinemia is not the only adverse effect of nitrites on fish. Nitrites also impair ion regulation, cardiovascular function, endocrine processes, and excretory mechanisms [77]. Exposure to acute or chronic sublethal nitrite concentrations increases fish susceptibility to infectious diseases, suggesting a negative impact of nitrites on the immune system.

Nitrites affect not only fish but also other aquatic organisms such as amphibians, crustaceans, mollusks, and algae. In amphibians, as in fish, methemoglobinemia has been observed, although the mechanisms of nitrite toxicity in these organisms are less well understood [78]. In frogs and their early developmental stages, nitrite toxicity primarily stems from its ability to penetrate the skin, disrupting respiration and osmoregulation. Among crustaceans, species such as crayfish and shrimp exhibit sensitivity to nitrites. In these organisms, nitrites disrupt osmoregulation, manifested as decreased chloride (Cl^−^) and sodium (Na^+^) ion concentrations in the hemolymph and reduced total osmolality of bodily fluids [79,80,81]. Nitrites also impair the immune systems of crustaceans, increasing their susceptibility to infections and diseases caused by pathogens [82,83]. In algae and cyanobacteria, nitrites can act both as a nitrogen source and as a toxicant. At low concentrations, algae and cyanobacteria can utilize nitrites as a nitrogen source when other substrates, such as ammonia or nitrates, are unavailable [84]. However, higher nitrite concentrations inhibit growth, induce oxidative stress, and damage photosynthetic processes [77].

Despite the negative impact of nitrogen forms on aquatic ecosystems, it is necessary to quantify the greenhouse effect potential of nitrogen oxides, which often escape the treatment process as by-products. Specifically, nitrous oxide (N_2_O), a significant nitrogen oxide emitted during wastewater treatment, has a global warming potential (GWP) approximately 265 times that of carbon dioxide (CO_2_) over a 100-year timeframe. This means that even small emissions of N_2_O can have a substantial impact on climate change [85]. While other nitrogen oxides like NO and NO_2_ also contribute to atmospheric chemistry, their direct GWP is less significant compared to N_2_O. However, they play a crucial role in the formation of ground-level ozone, another greenhouse gas, and contribute to acid rain.

### 1.4. Objectives and Scope of Review

The primary objective of this review is to comprehensively analyze advanced methods for nitrogen removal and recovery from municipal and industrial wastewater. The study seeks to address the growing environmental challenges associated with nitrogen pollution, including its contribution to eutrophication, greenhouse gas emissions, and water quality degradation. By examining both traditional and emerging technologies, the review aims to identify strategies that optimize nitrogen management in wastewater treatment processes. The scope of the review encompasses an in-depth exploration of biological processes, such as nitrification, denitrification, and the anammox process, alongside innovative approaches like shortcut nitrogen pathways. Additionally, the study evaluates physical and chemical methods, including ammonia stripping and membrane-based systems, with a focus on operational efficiency, economic feasibility, and environmental sustainability. This review also highlights critical factors influencing the performance of nitrogen removal systems, such as reactor configurations, process conditions, and external environmental impacts. The findings aim to provide a framework for improving wastewater treatment practices and promoting sustainable nitrogen management in line with evolving regulatory and environmental demands.

The literature analysis conducted in this review was based on a systematic and structured approach. Scientific databases including Scopus, Web of Science, ScienceDirect, and Google Scholar were used for collecting relevant peer-reviewed articles, book chapters, and technical reports. Initial searches returned more than 500 articles, from which approximately 230 publications were selected based on defined screening criteria, including relevance to municipal and industrial wastewater, methodological rigor, recency (priority given to the literature from the past 10 years), quality of experimental data, novelty of technological solutions, and publication in peer-reviewed journals. The selected references cover fundamental theoretical concepts, emerging technologies, and practical applications, providing a comprehensive overview of current knowledge and research trends in nitrogen removal and recovery.

## 2. Conventional Nitrogen Removal Methods

At the end of the 19th century, Sergei Winogradsky was the first to isolate chemolithotrophic bacteria capable of growth through nitrification, utilizing ammonia or nitrites as energy sources and electron donors. During his groundbreaking research, he discovered that nitrification occurs in two stages, each mediated by distinct groups of microorganisms: ammonia-oxidizing bacteria (AOB) and nitrite-oxidizing bacteria (NOB). Winogradsky observed that cultivating both groups of microorganisms under laboratory conditions required considerable time and patience [86].

### 2.1. The Nitrification Process: Mechanism and Environmental Conditions

Nitrification is a key step in the nitrogen cycle in the environment, involving the biological conversion of ammonium forms into nitrates. It essentially consists of two stages: the first involves the oxidation of ammonium ions to nitrites (NO_2_^−^), and the second involves the conversion of nitrites to nitrates (NO_3_^−^). The process occurs in two sequential steps. In the first stage of nitrification, chemolithotrophic bacteria (AOB) and archaea (ammonia-oxidizing archaea, AOA) convert ammonia into nitrites. This reaction requires the presence of oxygen as the electron acceptor.NH_4_^+^ + 1.5 O_2_ → NO_2_^−^ + 2 H^+^ + H_2_O [ΔG° = –275 kJ/mol], (2)

In the second stage of nitrification, chemolithotrophic bacteria (NOB) oxidize the resulting nitrites into nitrates.NO_2_^−^ + 0.5 O_2_ → NO_3_^−^ [ΔG° = –74 kJ/mol],(3)

The nitrification process is exergonic and thermodynamically spontaneous (ΔG^0^ < 0); however, it requires the supply of oxygen, which can be highly energy-intensive. Conventional wastewater treatment plants in Europe consume approximately 4.6 kWh per kilogram of ammonium nitrogen removed, translating to an annual cost of around half a million euros for a medium-sized plant (serving 200,000 population equivalents) [87]. In traditional wastewater treatment facilities, energy used for aeration accounts for about 60% of the total energy consumption in biological wastewater treatment processes [88]. However, there are methods to reduce energy consumption while maintaining nitrification efficiency, such as the application of micro-nano bubble technology, which can reduce aeration energy requirements by up to 50% [89].

#### 2.1.1. Nitrification Microorganisms

Both stages of nitrification are closely linked to the activity of specific groups of microorganisms. The energy acquisition process by AOB begins with the conversion of ammonia or ammonium ion into hydroxylamine, which is subsequently oxidized to nitrite. The enzymes ammonia monooxygenase (AMO) and hydroxylamine dehydrogenase (HAO) play critical roles in this process. AMO is composed of the amo operon, consisting of at least three subunits encoded by the genes *amoA*, *amoB*, and *amoC* [90]. With the advancement of molecular biology techniques in recent years, it was discovered that the *amoA* gene, a marker gene for ammonia oxidation, is also found in large quantities in archaea AOA [91]. The cell volumes of most AOA are 10 to 100 times smaller than those of known AOB. Consequently, the ammonia oxidation rate per single AOA cell is reported to be ten times lower than that of AOB [92]. On the other hand, AOA cell membranes, based on tetraether lipids, are less permeable to ions compared to the membranes of AOB. This reduces cyclic ion losses and lowers the energy required for cell maintenance, giving AOA an advantage in adapting to extreme environmental conditions [93].

Ammonia-oxidizing bacteria belong to the Proteobacteria phylum and are divided into two main groups: beta- and gamma-Proteobacteria. Among these, *Nitrosomonas*, *Nitrosospira*, *Nitrosovibrio*, and *Nitrosolobus* are classified as beta-Proteobacteria, while *Nitrosococcus* represents gamma-Proteobacteria [94]. Ammonia-oxidizing archaea are microorganisms that play a crucial role in the nitrogen cycle. Previously classified under the phylum *Thaumarchaeota*, recent taxonomic analyses have necessitated their reclassification into the class *Nitrososphaeria* within the phylum *Crenarchaeota*. All identified AOA belong to the order *Nitrososphaerales*, which includes diverse groups found in both marine and terrestrial ecosystems. At the genus level, key representatives include *Nitrosopumilus*, *Nitrososphaera*, *Nitrosotalea*, and *Nitrosocaldus*, each differing in environmental preferences and specific features related to ammonia oxidation. According to [95], AOA account for approximately 1–3% of all microorganisms in soils, with their abundance depending on environmental conditions. AOA prefer soils rich in organic matter and with low pH. Studies have shown that AOA can be found throughout the soil profile, in contrast to AOB, which typically prefer the topsoil layer [90].

Due to the toxicity of nitrites to eukaryotes and their inhibitory effect on bacterial growth, failures in their oxidation process can lead to significant ecological damage if nitrites escape from wastewater treatment plants into natural waters. Thus, the ability of nitrite-oxidizing bacteria to remove nitrites is crucial for environmental protection. Known bacteria involved in the second stage of nitrification belong to seven genera: *Nitrobacter*, *Nitrotoga*, *Nitrococcus*, *Nitrospira*, *Nitrospina*, *Nitrolancea*, and “Candidatus Nitromaritima” [96]. All NOB have cell envelopes typical of Gram-negative bacteria, except *Nitrolancea hollandica*, which stains as Gram-positive and forms thick cell wall layers [97]. The most diverse genus of NOB is *Nitrospira*, comprising at least six phylogenetic sublineages that are ubiquitous in nature [96]. Culturing NOB is challenging, with the isolation and purification process for some strains taking up to 12 years [98]. Techniques such as optical tweezers for sorting single cells [99] or separating cell aggregates via flow cytometry [100] facilitate the isolation of NOB, but generating sufficient biomass for further research often remains a challenge. For this reason, wastewater treatment plants are a natural source of NOB for study. For a long time, it was believed that the primary representatives of NOB in wastewater treatment plants were strains belonging to sublineages I and II of the *Nitrospira* genus. However, recent studies indicate that bacteria from the genus *Nitrotoga* also play a significant role in these systems. Interestingly, in some installations, both groups can coexist, sharing available ecological niches [96].

#### 2.1.2. Impact of Environmental Factors on Activity of Nitrifying Microorganisms

The growth kinetics of microorganisms involved in the first phase of nitrification is naturally dependent on the concentration of NH_4_^+^ in the environment. Studies indicate that AOA exhibit a higher affinity for NH_4_^+^ compared to AOB [91]. In practice, this means that AOA are more efficient at capturing and utilizing ammonia as a substrate at low concentrations. Research [101] conducted in three nitrifying bioreactors processing synthetic wastewater demonstrated that ammonia-oxidizing bacteria (AOB) were more competitive than ammonia-oxidizing archaea (AOA) across all tested ammonia concentrations (14, 56, and 140 mg/L NH_4_^+^-N). While AOA abundance remained relatively stable, AOB dominance increased with higher ammonia concentrations, indicating their greater role in nitrification under these conditions. Similarly, ref. [102] analyzed a bioreactor processing saline wastewater with ammonia concentrations up to 300 mg/L NH_4_^+^-N. Under these conditions, AOB, particularly from the genus Nitrosomonas, showed significantly higher abundance compared to AOA, which were present but marginal. In a study by [103], a full-scale landfill leachate treatment plant with exceptionally high ammonia concentrations, ranging from 440 to 2360 mg/L NH_4_^+^-N, was examined. In this system, AOA accounted for only 0.03–0.04% of the total microbial population. Considering the ammonium loading rate of the system, the effect of free ammonium [FA] must be taken into account. FA has a greater influence on the activity of NOB than on AOB, leading to nitrite accumulation due to its inhibitory effect on *Nitrobacter*. While FA concentrations in the range of 10–150 mg/L are required to decrease AOB activity, NOB inhibition occurs at a much lower range of 0.1–1 mg/L [104].

Another essential substrate in the nitrification process is oxygen, specifically dissolved oxygen (DO). Studies conducted as early as the 1970s indicate a wide range of DO requirements for nitrifying bacteria, varying from 0.3 mg/L to over 4 mg/L [105]. The primary factor contributing to the variability in results is the measurement conditions, encompassing both technical factors (e.g., reactor type) and technological factors (e.g., temperature). Under controlled laboratory conditions, DO concentrations above 1 mg/L do not significantly influence the growth of *Nitrosomonas* [105]. Similar effects are observed at constant temperatures in full-scale sequencing batch reactors (SBRs) [106]. In technical-scale applications (full-scale wastewater reactors), low DO levels (<1.3 mg/L) lead to a significant decline in nitrification activity during winter, despite the stable population of nitrifying bacteria [107]. To maintain efficient and complete nitrification in conventional wastewater treatment plants year-round, it is recommended to keep DO levels above 2 mg/L, particularly during the winter season [107].

Dissolved oxygen concentration also influences the activity and interactions among different NOB taxa. Variable aeration intensities in bioreactors can promote the growth of distinct *Nitrospira* sublineages, while environments such as biofilms or sediment layers create oxygen gradients that enable the coexistence of multiple NOB groups [108]. Particularly intriguing are observations from oxygen minimum zones (OMZs). Even at oxygen concentrations below 1 µM, significant nitrite oxidation activity, ranging from 36 to 59% of the levels observed under >10 µM oxygen, were recorded [109]. This suggests that some NOB have evolved exceptionally high oxygen affinity. Furthermore, it was hypothesized that these bacteria may use alternative electron acceptors under such conditions, including iodate, Mn(IV), or Fe(III) [110,111]. This metabolic flexibility contributes to the resilience and efficiency of nitrification processes in highly diverse environments.

For microorganisms involved in the first stage of nitrification, temperature primarily affects the activity of ammonia monooxygenase, the key enzyme in this process [112]. Currently known ammonia-oxidizing bacteria are mesophiles, with their temperature range typically limited to moderate environmental conditions [113]. In contrast, ammonia-oxidizing archaea exhibit a broad temperature adaptation range, spanning from cold oceanic environments to extremely hot geothermal springs, where temperatures can reach up to 70 °C [114]. This adaptability of AOA is attributed to the unique structure of their cell membranes, which contain glycerol ether lipids that provide high thermal stability [91]. The specificity of enzymatic activity highlights the competitive and environmentally dependent niches of AOA and AOB. Numerous studies indicate that under psychrophilic conditions (~15 °C), AOA dominate the nitrification process, even when AOB are numerically more abundant. Conversely, higher temperatures (up to 33 °C) favor the high activity of AOB [104,115,116,117]. This specificity allows nitrification processes to occur year-round in wastewater treatment plants (WWTPs) located at different latitudes.

Temperature is also a critical factor shaping the composition of NOB communities. *Nitrotoga* strains isolated from WWTPs thrive most efficiently at low temperatures (10–17 °C), similar to *Nitrotoga arctica* from permafrost soils [118,119]. Even at 27 °C, these bacteria retain some CO_2_ fixation activity, demonstrating their capacity to function in slightly warmer conditions. In contrast, *Nitrospira* strains exhibit a broader temperature tolerance range (10–28 °C) and gain a competitive advantage over *Nitrotoga* at higher temperatures [118]. An intriguing case is *Nitrotoga* with an optimal growth temperature of 22 °C and pH 6.8, compared to *Nitrospira defluvii* (sublineage I), which thrives at 32 °C and pH 7.3 [120]. These differences suggest that seasonal temperature fluctuations can promote NOB diversity in WWTPs, where cold-adapted strains are more active in winter, while heat-tolerant strains dominate in summer.

#### 2.1.3. Kinetic Modeling of Nitrification and Its Impact on Reactor Sizing

The efficiency of nitrification is strongly governed by process kinetics, which directly influences reactor design, sizing, and operational efficiency. Understanding these kinetics is essential for optimizing reactor parameters and ensuring effective nitrogen removal. Nitrification follows Monod kinetics, where the reaction rate is dependent on substrate concentration, microbial growth rate, and environmental conditions. The overall rate of ammonia and nitrite oxidation is governed by the specific growth rate (µ) of nitrifying bacteria, which is influenced by temperature, pH, dissolved oxygen levels, and substrate availability [121]. The process can be described using the Monod equation.(4)$$r=\frac{{µ}_{max}S}{K_{s}+S}$$
where
r = reaction rate.μ_max_ = maximum specific growth rate of nitrifying bacteria.S = substrate concentration (ammonium or nitrite).K_s_ = half-saturation constant (substrate concentration at which growth rate is half of μ_max_).

The growth rate of AOB is generally lower than that of heterotrophic bacteria, making nitrification a slower process. Additionally, NOB have an even lower growth rate, meaning that nitrite accumulation can occur under conditions where NOB activity is inhibited. Nitrification efficiency is highly sensitive to dissolved oxygen (DO) levels, as the process is strictly aerobic. The oxygen demand for nitrification is substantial, with approximately 4.57 mg O_2_ required per mg of NH_4_^+^ oxidized. If DO levels fall below 2.0 mg/L, nitrification rates decline significantly. Additionally, pH fluctuations impact bacterial activity, as nitrification produces hydrogen ions (H^+^), leading to acidification that may require alkalinity supplementation [122].

The kinetics of nitrification have a direct impact on reactor configuration, sizing, and operational parameters, particularly in biological wastewater treatment systems. Hydraulic retention time (HRT) must be carefully adjusted to allow sufficient contact time for microbial communities to oxidize ammonium efficiently. If HRT is too short, nitrification may be incomplete, resulting in elevated ammonia or nitrite concentrations in the effluent. On the other hand, excessively long HRT increases reactor volume, energy costs, and operational complexity [123]. Similarly, sludge retention time (SRT) plays a critical role in maintaining a stable population of nitrifying bacteria. Since AOB and NOB are slow-growing organisms, a longer SRT (typically >10 days) is required to prevent their washout from the system. In activated sludge processes, maintaining an adequate SRT ensures that the nitrifier population remains active and resilient to fluctuations in influent composition. The choice of reactor configuration also depends on nitrification kinetics. Suspended growth systems, such as conventional activated sludge or sequencing batch reactors (SBRs), require effective aeration and sludge management to support nitrification. In contrast, attached growth systems, such as biofilm reactors and moving bed biofilm reactors (MBBRs), offer advantages by retaining nitrifying bacteria on surfaces, allowing for higher microbial densities and increased resistance to shock loads [124]. In integrated fixed-film activated sludge (IFAS) systems, biofilm carriers provide additional surface area for nitrifiers, enhancing process stability.

As wastewater treatment systems transition from laboratory-scale research to full-scale applications, mathematical models and computational tools are used to predict nitrification performance and optimize reactor design. Traditional models, such as Monod kinetics and Michaelis–Menten equations, provide a basic understanding of microbial growth and reaction rates. More advanced models, such as the Activated Sludge Models (ASM1, ASM2, ASM3), incorporate nitrification along with carbon oxidation and denitrification processes, offering a more comprehensive framework for process optimization [125].

#### 2.1.4. Comammox

Nitrification has long been described as a two-step process. This division seemed natural, even necessary, since for over 100 years following the discovery of nitrifying bacteria by Sergei Winogradsky, no organism capable of performing both reactions simultaneously had been observed. However, this raised questions, as thermodynamic calculations clearly demonstrate that complete nitrification (oxidation of ammonia to nitrate in a single cycle) yields a higher energy gain (–349 kJ/mol NH_3_) compared to each individual step considered separately (–275 kJ/mol NH_3_ and –74 kJ/mol NO_2_^−^). Theoretically, a “complete” nitrifying bacterium—referred to as a comammox (complete ammonia oxidizer)—should exist, capable of conducting the entire process independently and thereby outcompeting “incomplete” nitrifiers. It was even hypothesized that such an organism would be particularly efficient in environments with limited substrate availability, such as biofilms. Yet, until recently, no known bacterium had confirmed this hypothesis, creating a significant paradox in nitrification research for many years.

This situation changed with the discovery of *Nitrospira* species capable of performing complete nitrification [126,127]. Unlike other known *Nitrospira* and all previously described organisms, these species possess a complete set of genes necessary for oxidizing both ammonia and nitrite. The first *Nitrospira comammox* strains were isolated from a trickling filter in an aquaculture system [127] and from a biofilm growing at moderately high temperatures at the bottom of an abandoned 1200 m deep oil well [126]. Subsequent metagenomic sequencing and analyses of new metagenomes revealed that genes characteristic of comammox are widely distributed in soil, freshwater, wastewater treatment plants, and drinking water treatment systems [96]. Furthermore, in one full-scale wastewater treatment plant, comammox was found to be the dominant ammonia oxidizer [126]. Combined with the phylogenetic placement of these bacteria within the common *Nitrospira* lineage II, this suggests that comammox organisms are typical nitrifiers in many environments and engineering systems [96].

The comammox process requires specific environmental conditions. Comammox bacteria (CAOB) are active under low dissolved oxygen (DO) levels, which may provide them with an advantage over other nitrifying microorganisms. At higher DO levels (e.g., 2–4 mg/L), nitrite accumulation is observed, suggesting that comammox activity may be inhibited under such conditions [128]. CAOB also exhibit greater stability and diversity at lower temperatures compared to conventional nitrifying bacteria. In a study [129] on the presence of CAOB in drinking water treatment systems across seasons, the most complex comammox community networks were observed in December. CAOB are most active within a narrow pH range of 7 to 8 [129].

### 2.2. Denitrification: Key Stages and Environmental Conditions

Denitrification is a microbiological process primarily carried out by heterotrophic bacteria. These microorganisms can utilize nitrates and nitrites as electron acceptors. Denitrification is a multi-step reduction reaction consisting of four main stages. In the first stage, nitrates are reduced to nitrites, a reaction catalyzed by nitrate reductase (NaR).(5)$${{\mathrm{NO}}_{3}}^{-}+2\;e^{-}+2\;H^{+}\;\overset{NaR}{\to }{{\mathrm{NO}}_{2}}^{-}+H_{2}O\;[\Delta G{}^{\circ}\approx -165\text{ kJ per reaction cycle}],$$
In the second stage, nitrites are reduced to nitric oxide, a reaction catalyzed by nitrite reductase (NiR).(6)$${{\mathrm{NO}}_{2}}^{-}+e^{-}+2\;H^{+}\;\overset{NiR}{\to }\mathrm{NO}+H_{2}O\;[\Delta G{}^{\circ}\approx -110\text{ kJ per reaction cycle}],$$
In the third stage, nitric oxide is reduced to nitrous oxide, a reaction catalyzed by nitric oxide reductase (NoR).(7)$$2\;\mathrm{NO}+2\;e^{-}+2\;H^{+}\;\overset{NoR}{\to }N_{2}O+H_{2}O\;[\Delta G{}^{\circ}\approx -300\text{ kJ per reaction cycle}],$$
In the final, fourth stage, nitrous oxide is reduced to molecular nitrogen, a reaction catalyzed by nitrous oxide reductase (NoS).(8)$$N_{2}O+2\;e^{-}+2\;H^{+}\;\overset{NoS}{\to }N_{2}+H_{2}O\;[\Delta G{}^{\circ}\approx -340\text{ kJ per reaction cycle}],$$

The enzymes involved in denitrification are produced when environmental conditions become favorable for this process. The synthesis of these enzymes is typically tightly regulated and is widely regarded as inducible. Their production mainly occurs under anaerobic conditions, although denitrification itself can also take place in the presence of oxygen. In certain cases, even low oxygen concentrations are required to initiate enzyme synthesis. Moreover, significant amounts of denitrification enzymes can be produced in anaerobic environments, even in the absence of nitrates or other nitrogen oxides [130]. Under controlled anaerobic or microaerophilic conditions, most organisms conducting denitrification strive for the complete reduction of nitrogen to its gaseous form (N_2_). However, in some cases, such as during suboptimal enzyme activity or intermittent oxygen availability, nitrous oxide (N_2_O) may be the final product. It is important to note that N_2_O is a potent greenhouse gas with a global warming potential approximately 300 times greater than CO_2_, and it also contributes to ozone layer depletion.

Most known denitrifying organisms belong to bacterial genera such as *Achromobacter*, *Acinetobacter*, *Aeromonas*, *Gallionella*, *Halobacterium*, *Halomonas*, *Hyphomicrobium*, *Janthinobacterium*, *Neisseria*, *Paracoccus* (formerly *Micrococcus*), *Propionibacterium*, *Pseudomonas*, *Rhizobium*, *Rhodobacter* (formerly *Rhodopseudomonas*), *Thiobacillus*, *Thiosphaera*, *Vibrio*, and *Xanthomonas* [130]. The denitrification process can also be carried out by certain autotrophic (chemolithotrophic) bacteria that use reduced sulfur or iron compounds as electron sources.

Key environmental factors affecting the activity of denitrifying bacteria (and thus the rate of denitrification in reactors) include pH, temperature, carbon source and concentration, and nitrate concentration. Optimizing denitrification requires maintaining a pH range of 7.5–8.0 and a temperature between 20 °C and 30 °C (e.g., [131]). At lower temperatures (10–15 °C), the denitrification rate decreases significantly, but the addition of redox mediators can improve process efficiency [132]. On the other hand, increasing the temperature can lead to greater accumulation of N_2_O during denitrification due to higher production rates and lower solubility in the liquid phase [133].

In contrast to autotrophic nitrifying bacteria, most heterotrophic denitrifying bacteria require a carbon source as an electron donor. The minimum C/N ratio necessary for effective denitrification depends on the characteristics of the treatment system. However, a low C/N ratio generally results in limited nitrogen removal and may lead to nitrite accumulation. Studies suggest that in biological reactors, the C/N ratio should be at least 2 to achieve stable and efficient denitrification. For example, at this ratio, bacterial strains such as *Pseudomonas stutzeri* can reduce nitrates with an efficiency of up to 92% [134]. Under more demanding conditions, a C/N ratio of 3–4 enables the removal of up to 87% of nitrates in sequencing batch reactors [135]. Maintaining an optimal C/N ratio may require the addition of an external carbon source. Various carbon sources are used in denitrification processes, depending on system specifics and local availability. Commonly used are simple organic compounds such as methanol, ethanol, or acetic acid, which are efficient and easy to dose. Organic materials from waste, such as molasses, whey, or distillation by-products, are also popular due to their low cost and availability, especially in systems integrated with the food industry. In some cases, natural organic substrates such as straw, tree bark, or cotton fibers serve as inexpensive and environmentally friendly alternatives. There is growing interest in biodegradable polymers, such as synthetic polyester granules, which slowly release carbon, making them particularly useful in long-term systems. Each of these carbon sources has its advantages and limitations, with the choice depending on cost, denitrification efficiency, ease of use, and the specific conditions of the treatment system [130]. Among these, methanol is especially preferred in research and engineering practice due to its relatively low cost, ease of dosing, and the ability to precisely control the process.

The denitrification process can be implemented in various types of treatment systems, categorized into fixed-film processes and suspended-growth processes. In fixed-film processes, anaerobic (anoxic) filters allow denitrification under low oxygen conditions, while rotating anaerobic disk filters provide a large surface area for microbial activity. Expanded bed granular reactors (upflow systems) involve water flowing upward through a filtration medium, enhancing contact and efficiency. Anaerobic fluidized bed systems use a filtration medium suspended by water flow, which increases the effectiveness of the process. In suspended-growth processes, denitrification can be carried out in separate sludge systems, where activated sludge is isolated from the denitrification process, or in integrated systems with anoxic zones. The latter configuration allows simultaneous denitrification and aerobic treatment within the same unit, achieved through alternating aerobic and anoxic zones. These configurations enable effective nitrogen removal and can be adapted to the specific requirements of the wastewater treatment system.

### 2.3. Summary of Biological Nitrification and Denitrification Processes

Biological nitrogen removal, encompassing the sequential processes of nitrification and denitrification, forms the core of conventional wastewater treatment methods. Both stages of nitrification are thermodynamically favorable but require significant energy input for aeration. The discovery of comammox bacteria, such as those in the *Nitrospira* genus capable of complete nitrification within a single organism, has expanded the traditional two-step perspective. In practice, nitrogen removal technologies through nitrification and denitrification are optimized to reduce energy consumption (e.g., by utilizing micro-nano bubbles) and to maintain stable system performance under variable environmental and operational conditions, as summarized in Table 2.

A crucial complement to nitrification is denitrification, primarily conducted by heterotrophic bacteria that use nitrates and nitrites as electron acceptors. This process involves multiple reduction stages—from nitrates to nitrites and ultimately to molecular nitrogen—and is regulated by a complex of enzymes induced under limited oxygen availability. Effective denitrification requires an adequate carbon source (e.g., methanol, ethanol, or waste-derived materials), optimal pH and temperature conditions, and a controlled C/N ratio to prevent the accumulation of intermediate nitrogen forms (e.g., N_2_O). Given the diversity of microorganisms and their specific requirements (e.g., oxygen concentration and temperature), environmental engineering employs various systems, including fixed-film and suspended-growth configurations, to effectively integrate aerobic and anaerobic processes. This approach ensures high nitrogen removal efficiency while simultaneously reducing energy costs and minimizing environmental impact. The fundamental pathways in the nitrogen cycle are presented schematically in Figure 2. The anammox process is discussed in detail in Section 3.2: Novel Biological Processes Based on Shortcut Nitrogen Removal.

## 3. Advanced Nitrogen Removal and Recovery Methods

### 3.1. Ammonia Stripping

Ammonia stripping is a widely used technique in environmental engineering, particularly in water and wastewater treatment processes. The method involves removing ammonia by facilitating its transfer from the liquid phase to the gas phase and expelling it from the reactor through mass transfer. Industrially, this is achieved by raising the solution’s pH to alkaline levels (>pK_a_ of NH_4_^+^/NH_3_) and forcibly removing free ammonia from the reactor using air.

Air stripping is the most common stripping method for removing ammonia from wastewater. The analysis of various technologies for ammonia removal and recovery across different wastewater types highlights both the versatility and challenges of this process. Ammonia removal efficiencies ranging from 91% to 99% depend on wastewater characteristics, pH adjustment methods, and reactor configurations. Commonly treated wastewaters include industrial effluents, landfill leachates, livestock wastewater, and post-digestion digestates. Popular technologies include packed towers and stripping columns. The most frequently used chemicals for pH adjustment are lime and NaOH, with optimal pH ranges from 10.0 to 12.0, which promote ammonia volatilization. Higher temperatures (up to 70 °C) and increased airflow rates (typically ranging from 0.12 to 0.9 m^3^/(h·L), depending on system size and operational scale) significantly enhance efficiency but also raise operational costs. Ammonia is typically captured using sulfuric acid (H_2_SO_4_), enabling the production of ammonium sulfate, a by-product commonly used as fertilizer. Despite its high efficiency, this process faces challenges such as the impact of gas composition on removal rates, costs associated with chemicals and operations, and scaling issues when treating wastewater with high solids content (TS > 5%) [138] (Table 3).

Another type of stripping is steam stripping. While both air stripping and steam stripping involve similar mass transfer processes, steam stripping is essentially a distillation process conducted at elevated temperatures, typically near the boiling point of water. A key advantage of steam stripping is that it eliminates the need for treating off-gases, as the steam from the exhaust can be condensed into a small volume of fluid containing concentrated ammonia [150]. This method demonstrates a distinct advantage over air stripping in removing compounds with low volatility compared to ammonia, such as nitrobenzene [151]. The simpler process structure eliminates the need for multi-stage gas treatment and neutralization, which are characteristic of the air stripping method. An economic analysis [152] revealed that the total costs of steam stripping are approximately 17% lower than those of air stripping, making this technology more cost-effective.

The efficiency of steam stripping depends on several operational parameters. One of the most critical factors affecting process effectiveness is the gas-to-liquid ratio (G/L). Higher G/L ratios improve ammonia transfer; however, excessively high values can lead to increased energy consumption.

The variety of reactor designs for steam stripping allows the technology to be adapted to specific wastewater types and operational requirements. Countercurrent columns, such as those used in the metallurgical industry, achieve ammonia removal efficiencies of up to 99.9%. Packed columns filled with materials that enhance the contact surface between phases are commonly used for wastewaters such as cattle manure or separately collected human urine. In these systems, efficiency ranges from 83% to 96.8%, depending on the applied gas-to-liquid ratio and other operational parameters.

An innovative approach to intensifying the process is the use of rotating packed beds (RPBs), which employ centrifugal forces to enhance mass transfer. These systems, applied to high ammonia concentrations (5000–20,000 mg/L), achieve removal efficiencies of 95–98.8% (Table 4).

When comparing steam stripping to other methods, such as air stripping, its clear advantages become evident. Removal efficiencies reaching up to 99% surpass those achieved in traditional air-based systems. Additionally, steam stripping enables the production of highly concentrated ammonia solutions (up to 22.88% by weight), increasing the value of the recovered material [150]. The compactness and versatility of systems like rotating packed beds further enhance their attractiveness for industrial applications. Despite its effectiveness, steam stripping is associated with certain challenges. Technical issues include scaling up the technology, problems with inorganic deposit formation, and foaming during ammonia removal. Furthermore, the recovered ammonia, often in the form of ammonia solution or ammonium sulfate, requires proper storage and utilization. In recent years, numerous innovations in ammonia stripping technologies were introduced to improve process efficiency, reduce operational costs, and minimize environmental impact. The introduction of systems like RPBs has reduced space requirements and improved energy efficiency, making the technology more competitive. Key approaches include the use of microwaves, external electric fields, high-gravity techniques, and solar-assisted systems. Each of these methods offers unique benefits and is suitable for different industrial conditions.

The application of microwave radiation enables rapid and uniform heating of aqueous solutions, significantly enhancing the efficiency of ammonia stripping. Studies have explored this innovative approach. In research [158] on the use of microwaves for ammonia removal from coke plant wastewater, an ammonia removal efficiency of 80% was achieved. Considering the system’s capacity (5 m^3^/day) and initial ammonia concentrations (up to 11,000 mg/L), the result is promising. A key factor was the ability of microwaves to evenly heat the solution, increasing stripping efficiency while minimizing energy losses. The authors highlighted that microwave technology is more energy-efficient than traditional evaporation methods, though its application is limited by the cost of technological components. In another study [159], microwave-assisted air stripping was optimized using advanced methods such as the Taguchi method. The results showed that parameters like airflow rate, temperature, mixing speed, and exposure time could be finely tuned to achieve near-complete ammonia removal. Using 500 W microwaves and a temperature of 60 °C, the process achieved ammonia removal efficiency close to 100%. The study emphasized that this technology not only enhances process efficiency but also allows for substantial energy savings compared to traditional methods. Further advancements were demonstrated by [160], who combined microwave heating with high-gravity technology, significantly improving ammonia recovery efficiency. Their experiments achieved a process efficiency of 99.3%, a highly satisfactory result for industrial applications. High efficiency was attributed to enhanced mass transfer in the liquid–gas system and the reduced size of technological equipment, lowering operational costs. This advanced technology proved particularly valuable in applications where space and process time are critical constraints.

Another innovation involves the use of external electric fields to enhance gas stripping processes. Research [161] showed that an electric field intensity of up to 15 V/cm significantly improved ammonia diffusion efficiency, especially under challenging conditions such as low temperatures or limited airflow. Importantly, this technology eliminates the need for additional solution heating, greatly reducing energy costs. The authors also highlighted the ecological benefits of the method, including reduced reliance on chemicals and a lower environmental impact.

Innovations also include the use of renewable energy, such as solar energy, to support the stripping process. Research by [162] focused on employing solar energy to enhance ammonia stripping. In their study, vacuum collectors were used to heat the solution to 45 °C, achieving an ammonia removal efficiency of 98%. Solar energy eliminated the need for electricity for heating, reducing operational costs and increasing the process’s environmental sustainability. The authors emphasized the potential application of this technology in regions with high solar exposure, where its efficiency could be even greater.

### 3.2. Novel Biological Processes Based on Shorcut in Nitrogen Removal

The discussion of the deammonification process (based on a shortcut nitrogen removal pathway) must include the anammox process, which is an integral component of this technology. The term anammox (anaerobic ammonium oxidation) refers to a biological process in which ammonium ions are oxidized by nitrite to produce molecular nitrogen. The anammox process was accidentally discovered in 1995 in a fluidized bed denitrification reactor. During prolonged reactor operation, a sudden disappearance of ammonium ions in the influent wastewater was observed, coinciding with nitrate reduction [137]. Anammox may account for up to 50% of molecular nitrogen production in marine environments, emphasizing its role in the natural nitrogen cycle [163].

The anammox reaction can be represented as follows:NH_4_^+^ + NO_2_^−^ → N_2_ + 2 H_2_O [ΔG° ≈ −358 kJ/mol], (9)

The anammox reaction generally occurs under anaerobic conditions and is an autotrophic process. Anammox can also coexist with other reactions in environments with very low oxygen concentrations (microaerophilic), such as in nitrification-anammox systems (CANON). In such conditions, oxygen is consumed by other microorganisms, while anammox occurs in anaerobic microenvironments. The generation time (i.e., the time required for the population to double) of anammox bacteria under optimal laboratory conditions is approximately 11 days, typically ranging from 2 to 3 weeks. Due to their exceptionally slow growth rate, the incubation period for enriched cultures of anammox bacteria in continuous reactors often extends to 100–200 days. Interestingly, the biomass yield of anammox bacteria, approximately 0.07 moles of carbon per mole of oxidized ammonium, is comparable to the yield of conventional nitrifying bacteria. The slow growth rate of anammox microorganisms primarily results from their low substrate conversion rate rather than low energy utilization efficiency.

Anammox bacteria are grouped into three main genera: *Brocadia* (including *B. anammoxidans* and *B. fulgida*), *Kuenenia* (with the species *K. stuttgartiensis*), and *Scalindua* (including *S. wagneri*, *S. brodae*, and *S. sorokinii*). Phylogenetic analyses have shown that all three form a monophyletic lineage deeply rooted in the *Planctomycetes* phylum. The uniformity in metabolism and similar cellular ultrastructure of these bacteria indicate that the ability to perform the anammox process evolved only once [164]. These anammox bacteria have specific environmental requirements, enabling the preferential enrichment of certain genera in bioreactor systems. For instance, high salinity in a bioreactor favors the enrichment of *Scalindua*, while the addition of acetate alongside ammonium, nitrites, and nitrates promotes the enrichment of *Brocadia fulgida* [165,166]. The optimal temperature for growth and activity of anammox bacteria in engineered systems generally ranges between 35 and 40 °C [167]. Recent research has also focused on utilizing the anammox process at low temperatures (5–15 °C), referred to as “cold anammox” [168]. In addition, the anammox process operates effectively across a wide range of ammonia concentrations, from low levels around 2 mg N/L, where inhibition may occur, to high levels such as 700–2200 mg NH_4_^+^-N/L, depending on the operating system and reactor configuration. These findings demonstrate that the process can be applied not only in sidestream systems but also in mainstream wastewater treatment systems, broadening its potential applications.

The application of the anammox process in engineering systems offers significant savings in both operational costs and environmental impact. The anammox process requires substantially less oxygen compared to conventional nitrification–denitrification processes, reducing energy consumption for aeration by up to 60%. Unlike denitrification, anammox does not require the addition of external carbon sources such as methanol or acetate, eliminating the costs associated with purchasing and dosing these substances [169]. Due to the absence of external carbon requirements and reduced oxygen demand, CO_2_ emissions are decreased by up to 90% compared to traditional methods [165]. The anammox process is characterized by low biomass yield, which reduces the volume of secondary sludge requiring management, resulting in additional cost savings. Its high nitrogen removal efficiency allows for the use of smaller reactors or the integration of anammox into existing systems, thereby lowering construction and retrofit costs.

Anammox is particularly effective in treating wastewater with high nitrogen concentrations (e.g., effluents from sludge dewatering processes). However, the process requires a source of nitrite. Nitrite availability can be achieved through two approaches: partial nitritation-anammox (PNA) or partial denitrification-anammox (PdNA) (Figure 3).

In practice, to achieve the required nitrogen removal efficiency, PNA and PdNA processes are often combined in full-scale applications. The high effectiveness of deammonification was confirmed for wastewater generated during the dewatering of digested sludge, landfill leachates, and industrial effluents [136,169,170,171,172,173]. The shortcut nitrogen removal pathway can also be integrated with pretreatment methods aimed at carbon recovery for further applications, such as in high-rate activated sludge systems (HRAS), chemically enhanced primary treatment, leachate energy recovery, or even direct anaerobic wastewater treatment [174,175]. However, full-scale implementations remain predominantly limited to sidestream flows, with only a few successful applications in mainstream systems reported to date [176].

### 3.3. Membrane Technologies

Membrane technologies for wastewater treatment are gaining increasing importance in the context of stricter environmental regulations and the growing demand for efficient methods of recovering nutrients. Depending on the type of membrane and process parameters (pressure, temperature, module type), targeted removal of specific contaminants, including nitrogen compounds, is possible. Each membrane is assigned a so-called performance rating, indicating its ability to reliably remove particles of a specified size or larger. Theoretically, the pore size of the active layer is a key parameter defining a membrane’s characteristics. However, accurately determining pore size has remained a significant challenge for decades, complicating filtration control and limiting the ability to precisely tailor membranes to specific process requirements. Well-established methods for membrane characterization include estimating molecular weight cutoff (MWCO), bubble point testing, water flux measurement, solute rejection testing, mercury porosimetry, vapor–liquid equilibrium, gas–liquid equilibrium or permporometry, solid–liquid equilibrium or thermoporometry, and microscopic techniques [177]. At the same time, the demand for technologies capable of the increasingly precise removal of small contaminants continues to grow, putting pressure on membrane manufacturers and driving the need for the development of new measurement methods to keep pace with advancements in filtration engineering.

The most commonly used membrane technologies include ultrafiltration (UF), nanofiltration (NF), reverse osmosis (RO), and membrane bioreactors (MBRs). Ultrafiltration is a separation method used for removing suspended solids, colloids, larger microorganisms, and dissolved macromolecules with sizes ranging from 1 to 100 nm [178]. However, ultrafiltration is not effective at retaining dissolved inorganic ions, including nitrogen ions. Nanofiltration lies between ultrafiltration and reverse osmosis in terms of separation capabilities. NF membranes have pore sizes ranging from 0.1 to 1 nm, corresponding to a molecular weight cutoff of 200–1000 Da, allowing them to effectively retain multivalent ions and soluble organic compounds with molecular weights above 200 Da [179]. This enables the removal of hardness, heavy metals, and larger organic compounds such as dyes, including nitrogen-containing dyes. However, NF membranes are not efficient at filtering monovalent ions, such as ammonium (NH_4_^+^), nitrite (NO_2_^−^), or nitrate (NO_3_^−^) ions. These compounds can only be effectively removed using reverse osmosis. RO membranes have pore sizes of approximately 0.1 nm, enabling the removal of monovalent ions and compounds with very low molecular weights, below 200 Da. However, a major challenge in this technology is membrane fouling, which can limit process efficiency. This issue is discussed in more detail later in this chapter. Membrane bioreactors (MBRs) combine biological wastewater treatment and membrane separation in a single system, enabling the efficient removal of nutrients, including nitrogen.

Due to their limited filtration capabilities, UF membranes are often used as a preliminary stage to remove larger particles before employing NF or RO. For instance, in the dairy industry, UF is used to concentrate proteins and lipids, which can then be utilized for biogas production [180]. UF is also an effective method for removing bacterial contaminants [181]. NF, with its ability to remove multivalent ions and lower energy consumption compared to RO, is frequently chosen for industrial wastewater treatment where extremely high purity standards are not required. For example, NF is widely applied in the textile industry for dye separation, demonstrating high retention efficiency [182]. NF is also used in water softening processes for industrial applications [181]. RO technology, initially designed for seawater desalination, has found extensive use in the treatment of municipal and industrial wastewater, including nitrogen-rich landfill leachates. RO systems employing disk-tube membranes, supported by prefiltration with sand and cartridge filters, demonstrate stable membrane permeability levels of 14–22 L/(m^2^·h) (LMH) and separation efficiencies ranging from 94% to 100% [183]. In a study by [184], RO systems showed exceptional performance, reducing COD levels by 98%, BOD_5_ by 99.7%, and nitrogen contaminants by 98%. Moreover, RO technology was highly effective in removing heavy metals, achieving removal efficiencies of 100% for mercury, 92% for nickel, 89% for lead, and 76% for cadmium.

In classical membrane processes, the dominant mechanism is the physical separation of molecules and ions due to the pressure difference across the membrane. The pressure required for filtration increases as membrane pore size decreases, which is directly related to the rise in filtration resistance. For example, in the study by [185], the optimal pressure for UF during the treatment of casing wastewater was 0.25 MPa (2.5 bar), achieving COD removal rates above 60% and BOD removal rates above 35%, with a permeate flux of 580 L/(m^2^·h). The application of NF required a pressure range of 1.4–1.6 MPa, achieving COD removal efficiency above 70% and BOD removal efficiency above 90%, with a permeate flux of 60 LMH. In [186], optimal pressures for UF and NF in textile wastewater treatment were reported as 2–7 bar and 4–15 bar, respectively. In another study [187], the nanofiltration process was conducted at transmembrane pressures of 12.5–18.5 bar. Reverse osmosis (RO) requires the highest pressures among these processes. The energy consumption of the RO process corresponds to the osmotic pressure multiplied by the volume of desalinated liquid.(10)$$E_{min}=\pi \cdot V,$$
where *E_min_* is the minimum energy consumption (e.g., in joules), *π* is the osmotic pressure (Pa), and *V* is the liquid volume (m^3^). The osmotic pressure of a liquid is essentially proportional to the molar concentration of ions and can be calculated using Van’t Hoff’s equation.(11)π=i·c·R·T,
where *i* is the Van ’t Hoff factor (dimensionless), representing the number of particles into which a solute dissociates, i.e., for NaCl i = 2; *c* is the molar concentration of solute (mol/m^3^); *R* is the universal gas constant (0.08206 L·atm/(mol·K)); and *T* is the absolute temperature (K). For simplicity, these calculations will be demonstrated using the example of seawater desalination. According to Equation (10), seawater with a salinity of 35 g/L (599 mol/m^3^) will have a theoretical *π* (osmotic pressure) of approximately 29.7 bar (at 25 °C). The theoretical minimum energy required to desalinate 1 m^3^ of seawater is 2.97 MJ (approximately 0.825 kWh). However, this represents the thermodynamic minimum—in practice, energy consumption observed in pilot studies reaches a minimum of 1.6 kWh/m^3^ [188]. According to calculations by [189], additional energy is consumed to overcome flow resistance through membranes (24% of specific energy consumption, SEC) and losses associated with the imperfect operation of energy recovery devices (ERDs; 20% SEC). Membrane resistance arises from limited permeability and the need to maintain sufficient pressure to drive water through the membrane. Meanwhile, energy losses in ERDs are due to their efficiency, which never reaches 100% due to mechanical and hydraulic phenomena. The simplest model describing the transport of mass, salts, and solvents through RO membranes is the Kedem–Katchalsky equation [177].(12)$$J_{v}=L_{p}\cdot \left(∆P-\sigma ∆\pi \right),$$
where *Jv* is the water flux rate through the membrane (m^3^/m^2^/s = m/s), *Lp* is the hydraulic permeability coefficient of the membrane (a constant depending on the membrane’s properties and temperature; m/s/bar), *ΔP* is the pressure difference across the membrane, *σ* is the membrane’s osmotic rejection coefficient (0 ≤ σ ≤ 1), and *Δπ* is the difference in osmotic pressure across the membrane. Since the goal of most early studies was seawater desalination, membranes with an osmotic rejection coefficient (*σ*) of 0.993 were sought. The first membrane material meeting these criteria was cellulose acetate. Currently, composite membranes made by interfacial polymerization technology have higher values for both *Lp* and *σ*. However, cellulose acetate membranes still occupy a small segment of the market due to ease of production, high mechanical strength, and resistance to degradation caused by chlorine and other oxidants, which is a problem for composite membranes. On the other hand, cellulose membranes are highly sensitive to the degree of acetylation of the polymer used in their production [190]. Since the discovery that RO membranes with high values of *Lp* and *σ* can be produced by interfacial polymerization, this method has become a new standard in the industry. Currently, composite membranes produced by this technology, such as the cross-linked polyester Toray PEC-1000, feature σ as high as 0.9995 [190].

One of the most significant limitations in RO processes is membrane fouling, caused by pore blockage and adsorption of dissolved substances onto the membrane surface [191]. The most common causes of membrane fouling include biofilm development [192,193], scaling, the presence of dissolved organic substances, and particles and colloids forming compacted deposits [194]. The formation of biofilm occurs when microorganisms adhere to the membrane surface and grow in the presence of a polysaccharide matrix [195]. To inhibit the growth of microorganisms, chlorine is often used, but RO membranes are sensitive to its free form, while they show much higher tolerance to chloramines—this is why chloramines are commonly used to limit microbial growth [177]. A key method to extend membrane life and reduce contamination is proper pretreatment of the feed liquid (e.g., leachate from landfills). Typically, this includes chemical coagulation, multi-stage filtration (e.g., ultrafiltration or microfiltration), actions to prevent the precipitation of solids (e.g., softening), and pH regulation through acidification.

One of the most perspective approaches is the linking of thermophilic systems with membranes. These systems combine the advantages of high-temperature biological processes and membrane separation technology to enhance treatment efficiency and resource recovery. Compared to conventional mesophilic biological nitrogen removal (BNR) processes, the thermophilic-ultrafiltration system presents distinct advantages and challenges (Table 5). By concentrating nitrogen-rich compounds in the retained sludge, the system facilitates downstream recovery techniques such as struvite precipitation or ammonia stripping. The membrane application provides a physical barrier to suspended solids and pathogenic microorganisms, improving effluent quality and making the treated water suitable for reuse applications. Moreover, the membrane effectively retains thermophilic microbial populations within the bioreactor, promoting higher sludge retention times (SRTs) and increasing process stability [196].

While thermophilic systems require additional energy input for heating, this can be offset by the potential for energy recovery (e.g., from anaerobic digestion) and the reduction in sludge disposal costs. Moreover, the integration with membrane processes significantly enhances process stability and effluent quality, making the system viable for applications in high-strength wastewater treatment (e.g., industrial effluents, landfill leachate, or livestock manure) [197].

Membrane bioreactors (MBRs) are an advanced technology combining biological processes with membrane separation, enabling wastewater treatment at levels not achievable by conventional methods. These systems offer, among other things, better-treated water quality due to effective removal of nutrients and pathogens, and compact system sizes. The history of MBRs dates back to the late 20th century, when the first systems were developed, initially based on solid–liquid separation, and later submerged membrane bioreactors, which are now dominant in industrial applications. In 2010, the global MBR market reached USD 216 million, and by 2018, it was projected to grow to over USD 628 million, especially in regions with water shortages such as China and the Middle East. However, this technology faces challenges such as high energy consumption (approximately 0.5 kWh/m^3^ of liquid) and membrane fouling [198]. MBR technology relies on the integration of conventional biological processes with membrane filtration techniques. Typical membrane permeability values in MBR systems range from 10 to 150 LMH, which ensures high efficiency with minimal pressure loss. Furthermore, due to technological advancements, energy demand has decreased from 3 to 6 kWh/m^3^ in the 1960s to current values, making MBR an attractive solution both economically and environmentally [198].

### 3.4. Electrochemical Methods

In recent decades, there has been increasing interest in technologies that allow for the efficient and rapid removal of nitrogen compounds. Electrochemical methods are one such alternative. By precisely controlling the electrode potential, mass transport processes, and selecting appropriate electrode materials, ammonia oxidation (at the anode surface) and/or nitrate and nitrite reduction (at the cathode surface) can be conducted in a controlled manner. The main advantages of electrochemical treatment include high efficiency, the ability to operate at ambient temperature, compact device sizes, minimal sludge production, and a fast process startup.

During the electrochemical reduction in nitrates, water electrolysis plays a key role, leading to the formation of hydrogen (at the cathode) and oxygen (at the anode) in the reaction environment.H_2_O + 2 e^−^ → H_2_ + 2 OH^−^,(13)4 OH^−^ → O_2_ + 2 H_2_O + 4 e^−^,(14)

Hydrogen produced at the cathode serves as an electron donor, which is crucial for the reduction in nitrates to nitrogen forms with lower oxidation states. Under ideal conditions, where the goal is the complete reduction in nitrate ions to molecular nitrogen (N_2_), the mechanism of nitrate reduction is as follows:NO_3_^−^ + 3 H_2_O + 5 e^−^ → ½ N_2_ + 6 OH^−^,(15)

Generally, however, the reaction is more complex and occurs in several steps.NO_3_^−^ + H_2_O + 2 e^−^ → NO_2_^−^ + 2 OH^−^,(16)NO_2_^−^ + 2 H_2_O + 3 e^−^ → ½ N_2_ + 4 OH^−^,(17)

This mechanism unfortunately causes nitrites to participate in intermediate reactions, which carries the risk of producing unwanted by-products, including ammonia and hydroxylamine.NO_2_^−^ + 5 H_2_O + 6 e^−^ → NH_3_ + 7 OH^−^,(18)NO_2_^−^ + 4 H_2_O + 4 e^−^ → NH_2_OH + 5 OH^−^,(19)

Ammonia accumulation is a common problem in the electrochemical reduction in nitrates. To mitigate the accumulation of ammonia and nitrites, chlorides are often introduced into the reaction solution, usually in the form of sodium chloride (NaCl). This process relies on the electrochemical oxidation of chloride ions (Cl^−^) at the anode, leading to the formation of chlorine molecules (Cl_2_), which further produce hypochlorous acid (HOCl) and hypochlorite ions (OCl^−^). Both HOCl and OCl^−^ exhibit strong oxidizing properties, which help prevent the accumulation of ammonia and nitrites.

An additional problem, as observed from reactions (15–17), is the gradual increase in pH around the cathode during the electrochemical reaction. This pH rise, besides potentially influencing ammonia accumulation, promotes the precipitation of magnesium and calcium ions as hydroxides (Mg(OH)_2_) and carbonates (CaCO_3_). These compounds tend to deposit on the electrode surface, gradually hindering the electrochemical process. To stabilize the pH in the solution, sodium bicarbonate (NaHCO_3_) is often used.

Operational parameters that significantly influence the electrochemical denitrification processes include the electrode material, current density, pH, conductivity, sodium chloride (NaCl) concentration, and the initial concentrations of ammonium nitrogen and nitrate ions. The optimal NaCl concentration for electrochemical denitrification lies in the range of 0.5–1 g/L [199], and the pH should be stabilized between 7.0 and 8.5 to minimize the accumulation of by-products. In practice, various solutions are tested to improve the efficiency of electrochemical nitrate reduction. Research on electrode materials shows that different types of materials affect the selectivity and efficiency of the process. For example, Cu–Zn electrodes (62.2% Cu, 37.8% Zn) used in undivided reactors achieved nitrate reduction from 100 mg/L to 9.7 mg/L in 300 min at a current of 40 mA/cm^2^ and neutral pH. On the other hand, BDD electrodes allowed the removal of nitrates to levels below 50 mg/L in agro-industrial wastewater (2.2 g/L NO_3_^−^), at a current density of 300 A/m^2^ and pH 1.5, with an energy consumption of 25 kWh/kg NO_3_^−^. In alkaline conditions, Cu (100) electrodes were more effective in reducing nitrates to hydroxylamine (NH_2_OH), while Cu (111) favored the formation of nitrites (NO_2_^−^).

Optimization of process conditions includes adjusting voltage and current density. For example, studies with asymmetric voltage pulses (−1.7 V for 80% of the cycle and +1.5 V for 20% of the cycle, frequency 30–50 Hz) showed a nearly 100% nitrate reduction with minimal formation of ammonia and nitrites. Stabilization of pH at 6.5–7.0 in bioelectrochemical processes allowed for the removal of 96% of nitrates from an initial concentration of 200 mg/L NO_3_^−^–N in 5 h at a voltage of −0.6 V.

Reactor design solutions are also crucial. Two-chamber reactors with ion-exchange membranes, such as those used with BDD electrodes, allow for the separation of anodic and cathodic products, reducing the risk of undesirable products like ammonia. In tests with synthetic wastewater (27 g/L NO_3_^−^) using a Nafion 117 membrane reactor, 89% nitrate removal was achieved at a current of 2500 A/m^2^ and a temperature of 34 °C (Table 6).

The electrochemical oxidation of ammonia is also a multi-step process. In this process, the NH_3_ molecule undergoes gradual dehydrogenation on the electrode surface, and the resulting active nitrogen fragments can react with each other in various ways. The complexity arises from competitive reaction pathways, the presence of various intermediate products, and the fact that a slight change in electrode potential or experimental conditions (pH, temperature, NH_3_ concentration) can significantly shift the equilibrium towards the formation of different nitrogen compounds—ranging from molecular nitrogen to nitrates, and even nitrogen oxides (NO, N_2_O). An additional complication is the strong dependence of the mechanism on the microscopic properties of the electrode material surface. For example, platinum may be covered to varying degrees with oxides (PtO_x_), which alter its activity and selectivity, and may also adsorb some intermediates (e.g., NH_2_ or N(ads)), leading to “surface poisoning” [206].

In early studies (1963), it was assumed that the oxidation of ammonia occurs analogously to its decomposition in the gas phase. The proposed mechanism involved successive dehydrogenation steps of ammonia, culminating in the direct recombination of atomic nitrogen (N(ads)) into molecular nitrogen (N_2_) [207]. These steps can be presented as follows:NH_3_ + OH^−^ → NH_2_(ads) + H_2_O + e^−^,(20)NH_2_(ads) + OH^−^→ NH(ads) + H_2_O + e^−^,(21)NH(ads) + OH^−^→ N(ads) + H_2_O + e^−^,(22)2N(asds) → N_2_,(23)

However, further research showed that N(ads) is not the dominant step leading to the formation of N_2_, as atomic nitrogen can strongly adsorb onto the electrode surface, leading to the blocking of its catalytic activity. Instead, a mechanism was proposed in which the key step is the dimerization of NH_2_(ads) or NH(ads) fragments, leading to the formation of nitrogen dimer compounds (N_2_Hy) [208]. These steps can be written as follows:NH_2_(ads) + NH(ads) → ½ N_2_H_y_(ads),(24)N_2_H_y_(ads) + yOH^−^→ N_2_ + yH_2_O + ye^−^,(25)

The most common intermediate in this pathway is hydrazine (N_2_H_4_), which then oxidizes to molecular nitrogen.N_2_H_4_ + 4 OH^−^→ N_2_ + 4 H_2_O + 4 e^−^,(26)

Studies [208] have confirmed that the N–N bond can form without the need for atomic nitrogen and that surface reactions on the electrode, specifically between NH_2_(ads) and NH(ads), are crucial. These findings were supported by differential electrochemical mass spectrometry (DEMS), which confirmed the presence of N_2_ as the main product at low anodic potentials. At higher potentials, nitrogen oxides and overoxidation products such as nitrites and nitrates also appear.

In summary, the most probable mechanism of electrochemical ammonia oxidation involves the stepwise dehydrogenation of ammonia into NH_2_(ads) and NH(ads) fragments, which then condense into dimeric compounds like N_2_Hy, eventually oxidizing into molecular nitrogen. These insights have enabled a better understanding of the role of intermediates and potential reaction pathways depending on experimental conditions and electrode potential.

Research on electrochemical ammonia oxidation focuses on optimizing electrode materials, process parameters, and reactor designs to enhance the technology’s efficiency in wastewater treatment. A key aspect is the selection of suitable anode and cathode materials. For instance, Ti/PbO_2_ [209] and Ti/BDD [210] electrodes have demonstrated high efficiency in ammonia removal due to their stability and catalytic activity. An innovative application of the Ti/RuO_2_-Pt electrode [211] achieves an NH_3_-N reduction of 88.3% in just 30 min, making this technology attractive for practical applications.

Process parameters are critical for ammonia removal efficiency. The optimal current density varies depending on the type of wastewater, ranging from 20 mA/cm^2^ [211] for municipal wastewater to 90 mA/cm^2^ [212] for wastewater with high ammonia concentrations. The solution’s pH also plays an important role, with neutral or moderate alkaline conditions being the most effective due to the chloride cycle and efficient indirect oxidation. Additives like NaCl, at concentrations from 0.5 g/L to 8 g/L, enhance performance by generating active chlorine, which is the primary oxidizing agent in many studies.

The design of electrochemical reactors also affects results. For example, a study [213] introduced an innovative floating cathode concept, enabling ammonia recovery during the process. Most studies have used reactors with small capacities (0.5–0.57 L) and electrode spacing ranging from 1 to 40 mm, allowing for the precise control of the electric field and optimization of energy costs.

Oxidation mechanisms are dominated by an indirect process where active chlorine (Cl_2_, ClO^−^) oxidizes ammonia to molecular nitrogen. The direct mechanism plays a minor role, particularly on electrodes like Ti/RuO_2_-Pt or Ti/BDD. Optimizing these mechanisms, combined with energy efficiency research, can significantly reduce operational costs. Energy costs range from 0.16 kWh/g of NH_3_ [209] to 41–65 kWh/kg of COD [210], highlighting the need for further advancements (Table 7).

### 3.5. Technology Readiness and Scalability of Nitrogen Removal and Recovery Methods

Modern methods of nitrogen removal and recovery from water and wastewater differ not only in their mechanisms of action but also in their level of technological maturity (Technology Readiness Level—TRL) and potential for industrial-scale applications. This chapter discusses four key groups of nitrogen removal and recovery methods. Each described technology is characterized by a different level of operational complexity, operating costs, and infrastructure requirements, which influence its readiness for implementation and scalability.

Ammonia stripping (i.e., its transfer from the liquid phase to the gas phase) is one of the longest-used physicochemical methods for reducing nitrogen concentrations in liquid phase. The process typically employs towers or packed columns, where the wastewater stream is contacted with air or steam under alkaline pH conditions, facilitating the transition of ammonium nitrogen into NH_3_. This technology is widely documented in the literature and engineering practice, and its TRL can be considered relatively high (8–9), given the numerous municipal and industrial installations operating worldwide. This indicates significant technological maturity, although efficient operation requires optimization of pH, temperature, and gas flow parameters. The main advantage of ammonia stripping is its ability to treat a wide range of wastewater streams, from municipal to highly concentrated industrial effluents. The method is relatively easy to scale: it can be implemented in both small systems (e.g., pilot columns treating tens of cubic meters per day) and large-scale installations (towers in wastewater treatment plants serving hundreds of thousands of PE). Ammonium stripping is a widely used technique for removing ammonia from wastewater; however, it comes with several limitations. One of the primary challenges is its high energy demand, as the process requires significant heating and aeration to achieve effective ammonia removal. Additionally, pH dependence is a critical factor—ammonium stripping operates optimally at a pH above 10, necessitating the addition of alkaline chemicals, which increases operational costs and process complexity [150]. Another limitation is the potential loss of ammonia to the atmosphere, which can contribute to air pollution if proper recovery or scrubbing systems are not in place. Furthermore, ammonium stripping is less effective at low ammonium concentrations, making it unsuitable for certain wastewater streams where ammonia levels are insufficient for efficient removal. Scaling and fouling issues also pose operational challenges. At high pH, minerals such as calcium carbonate can precipitate, leading to scaling in stripping towers and requiring regular maintenance. These factors must be carefully considered when evaluating the feasibility of ammonium stripping in wastewater treatment applications [121,214].

In the past two decades, groundbreaking biological nitrogen removal processes based on shortcut pathways—bypassing the full nitrification–denitrification cycle—have been introduced. In both municipal and industrial applications, deammonification involving anammox is considered the most advanced method, allowing for a significant reduction in oxygen demand (by up to 60%) and carbon source consumption compared to conventional nitrification–denitrification. Despite its high potential for reducing operating costs, anammox-based methods are still implemented in a limited number of full-scale installations, particularly for wastewater with lower temperatures or variable compositions. According to market reviews, the TRL of these technologies ranges from 6 to 8, depending on the stability of full-load operation. The main challenge in scaling shortcut nitrogen removal processes is the long startup time of the bioreactor, resulting from the slow growth of anammox bacteria. Other critical factors include microbial sensitivity to pH fluctuations, temperature changes, and toxic substances in wastewater. Nevertheless, several reference installations successfully apply deammonification, e.g., for treating digestate with a high nitrogen load.

Membrane processes (e.g., gas membrane distillation, reverse osmosis, nanofiltration) are increasingly used for nitrogen removal, particularly for highly concentrated wastewater or water reuse applications. In nitrogen removal, key systems include those where ammonium nitrogen can be transported across a membrane in gaseous form (e.g., ammonia membrane distillation) or where nitrogen compounds are separated as ions (RO, NF). Although these technologies are often in an advanced stage of development (TRL 7–9), they can still be costly for high-flow applications. Membranes, particularly in high-pressure processes, require careful operation to address issues such as fouling, sensitivity to colloidal particles, and the need for periodic regeneration. Membrane technologies can be implemented in modular systems, facilitating their expansion as demand grows. Major barriers include high sensitivity to organic contaminants (biofouling), which necessitates precise pretreatment of wastewater. Additionally, energy costs, particularly in pressure-driven systems (NF, RO), increase with flow rates, limiting competitiveness in certain applications compared to traditional techniques. Nevertheless, membrane technologies play a key role in water recirculation and nutrient recovery.

The last group of nitrogen removal methods includes electrochemical techniques, where redox reactions (reduction or oxidation of nitrogen compounds) occur on electrode surfaces. These methods allow for the removal of both nitrates (NO_3_^−^)—through multi-step reduction to N_2_—and ammonia (NH_3_)—via electrochemical oxidation. Many systems utilize chloride ions (Cl^−^) to generate active chlorine (Cl_2_, HOCl, OCl^−^), which aids in converting intermediate forms to N_2_. Depending on reactor design (single- or dual-chamber) and electrode materials (e.g., PbO_2_, BDD, graphite), nitrogen removal efficiencies of up to 90–100% can be achieved at laboratory and pilot scales. However, at larger scales (high wastewater flow rates), this technology still requires optimization in terms of energy costs and long-term electrode stability (passivation, deposit formation). As a result, the TRL of electrochemical nitrogen removal methods is generally around 5, although there are examples of commercial applications in industrial sectors.

### 3.6. Fate of Nitrogen Within Biological Sludge Treatment

As wastewater treatment shifts toward sustainable and circular economy approaches, the fate of nitrogen within biological sludge has become a critical consideration. Traditional wastewater treatment primarily focuses on nitrogen removal through biological processes such as nitrification–denitrification, anaerobic ammonium oxidation (anammox), and assimilation into biomass. However, the residual nitrogen embedded within biological sludge poses a challenge for resource recovery efforts, particularly when sludge minimization techniques, such as incineration, lead to significant nitrogen losses into the atmosphere. Understanding the transformations and pathways of nitrogen within sludge is essential for optimizing treatment strategies and maximizing nitrogen recovery.

Biological sludge, a by-product of wastewater treatment, is primarily composed of microbial biomass, extracellular polymeric substances (EPSs), and accumulated organic and inorganic compounds. The fate of nitrogen in sludge depends on various factors, including microbial activity, sludge stabilization methods, and final sludge disposal or recovery processes. Sludge minimization techniques aim to reduce the volume of sludge generated, lower disposal costs, and enhance energy recovery. However, these processes also affect nitrogen fate, often leading to irreversible nitrogen losses. For instance, combustion of sludge at temperatures above 850 °C leads to almost complete oxidation of nitrogen into gaseous forms, predominantly nitrogen oxides (NO_x_) and molecular nitrogen (N_2_) [215]. While 95–99% of sludge nitrogen is lost during incineration, only a small fraction can be recovered through NO_x_ control systems (e.g., selective catalytic reduction, SCR) or as ash-bound nitrogen.

On the other hand, other thermal methods, such as sludge pyrolysis and gasification, which operate at lower temperatures (400–700 °C), enable the retention of a fraction of nitrogen in biochar, which can be utilized as a slow-release fertilizer [216]. Currently, for the majority of average and large WWTPs, anaerobic digestion (AD) is the most commonly applied sludge minimization technique. In the AD process, 20–40% of the sludge nitrogen is converted into ammonium, which accumulates in the digestate and requires post-treatment [217]. Ammonia recovery can be achieved through ammonia stripping, ion exchange, or membrane contactors. Nitrogen-rich digestate can be further processed for struvite precipitation, yielding a valuable phosphate- and nitrogen-based fertilizer.

Given the increasing global demand for nitrogen-based fertilizers and the environmental impact of nitrogen losses, future wastewater treatment must transition toward closed-loop nitrogen management. This can be achieved by integrating nitrogen recovery with renewable energy production (e.g., coupling anaerobic digestion with ammonia stripping), developing novel bio-based nitrogen fertilizers from wastewater treatment residuals, and optimizing sludge treatment to minimize nitrogen loss while maximizing energy and resource recovery.

## 4. Conclusions

The effective management of nitrogen in wastewater treatment is critical to mitigating its environmental impacts, including eutrophication, greenhouse gas emissions, and contamination of water resources. This review highlights both established and emerging methods for nitrogen removal and recovery, focusing on their operational principles, efficiency, and potential for sustainable implementation.

Conventional biological processes, such as nitrification and denitrification, remain the cornerstone of nitrogen removal in municipal and industrial wastewater treatment. However, innovative approaches, such as the anammox process and shortcut nitrogen pathways, offer significant advantages, including reduced energy demand, lower operational costs, and decreased greenhouse gas emissions. These advancements, combined with chemical and physical methods like ammonia stripping and membrane technologies, provide a versatile toolkit for addressing diverse wastewater characteristics.

To achieve optimal performance, the integration of advanced technologies into existing treatment systems requires careful consideration of factors such as reactor design, process conditions, and environmental constraints. The adoption of these solutions can not only improve nitrogen removal efficiency but also enable the recovery of valuable nitrogen-based products, contributing to a circular economy. Further research and development are essential to scale up these methods, address challenges in mainstream applications, and adapt to evolving regulatory and environmental demands.

By bridging traditional and innovative approaches, the wastewater treatment sector can achieve enhanced nitrogen management while promoting sustainability and resilience in the face of global environmental challenges.

## Figures and Tables

**Figure 1 materials-18-01422-f001:**
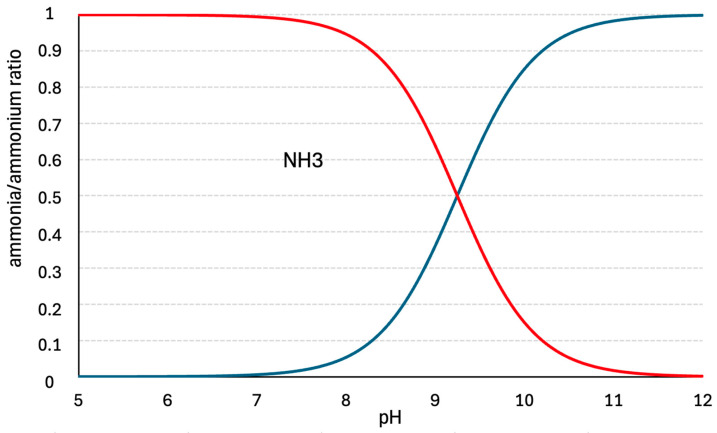
The NH_3_/NH_4_^+^ equilibrium as a function of pH in an aquatic environment at a temperature of 25 °C.

**Figure 2 materials-18-01422-f002:**
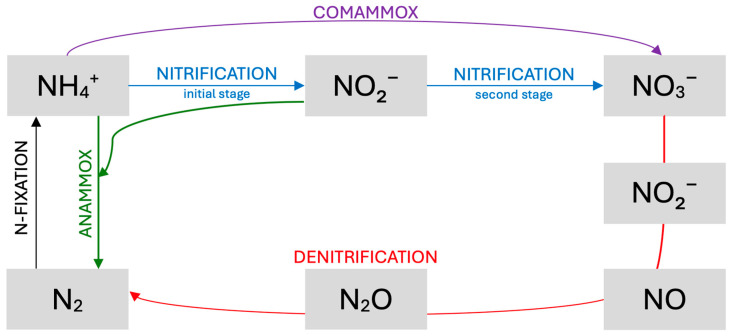
Basic pathways in nitrogen cycle. To enhance clarity, each pathway and its description are marked with different colors.

**Figure 3 materials-18-01422-f003:**
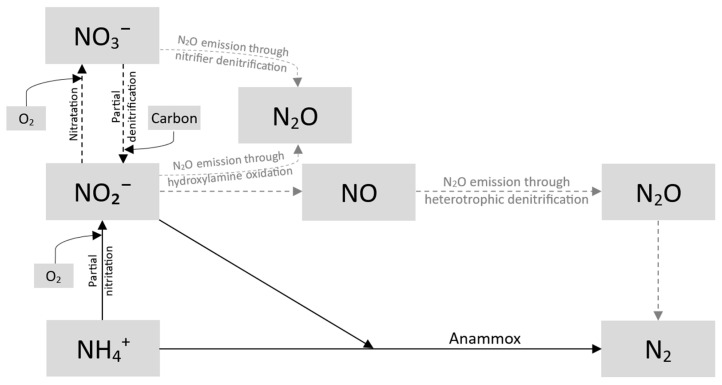
A scheme of deammonification encompassing partial nitritation-anammox and partial denitrification-anammox, including potential pathways for nitrous oxide emissions.

**Table 1 materials-18-01422-t001:** Typical concentrations of the nitrogen species in municipal wastewater (based on [6]).

Nitrogen Species	Typical Concentration Range (mg/L)	Remarks
Total Nitrogen (TN)	20–85	Sum of all nitrogen species, including organic and inorganic forms.
Organic Nitrogen	8–35	Includes proteins, amino acids, urea, and humic substances.
Ammonium (NH_4_^+^)	12–50	Predominant form of nitrogen in wastewater due to urea hydrolysis.
Nitrite (NO_2_^−^)	<0.1–0.5	Intermediate oxidation product, usually present in very low concentrations.
Nitrate (NO_3_^−^)	<0.1–1.0	Minimal in raw wastewater, but increases after nitrification in treatment processes.
Dissolved Organic Nitrogen (DON)	1–10	Comprises low-molecular-weight organic nitrogen compounds.

**Table 2 materials-18-01422-t002:** Main factors influencing nitrification and denitrification [19,54,105,128,131,136,137].

Factor	Influence on Nitrification	Influence on Denitrification
Dissolved Oxygen (DO)	Requires aerobic conditions (high DO). Inhibited by low DO.	Requires anoxic/anaerobic conditions (low or no DO). Inhibited by high DO.
pH	Optimal range: 7.0–8.5. Inhibited by very low or high pH.	Optimal range: 6.5–8.0. Can occur at a wider range but is generally slowed outside this range.
Temperature	Increases with temperature up to an optimal range (25–35 °C). Decreases at very low or high temperatures.	Increases with temperature up to an optimal range (30–35 °C.) Decreases at very low or high temperatures.
Carbon Source	Requires an inorganic carbon source (CO_2_, bicarbonates).	Requires an organic carbon source (e.g., BOD, COD).
Ammonium (NH_4_^+^) Concentration	Substrate availability. High concentrations may be inhibitory.	Generally, not directly relevant, but nitrate availability is crucial.
Nitrate (NO_3_^−^) Concentration	Product of nitrification.	Substrate availability. High concentrations may be inhibitory.
Inhibitory Substances	Heavy metals, sulfides, certain organic compounds.	Heavy metals, sulfides, certain organic compounds, high nitrite concentrations.
Hydraulic Retention Time (HRT)	Sufficient HRT is necessary for bacterial growth and reaction completion.	Sufficient HRT is necessary for bacterial growth and reaction completion.
Alkalinity	Consumed during nitrification.	Produced during denitrification.

**Table 3 materials-18-01422-t003:** Parameters and efficiency of air stripping for ammonia removal from various wastewater types.

Wastewater Type	Ammonia Concentr. [mg/L]	pH Adjuster	Reactor Type	Operational Conditions			$\eta_{{NH}_{3}}$	NH_3_ Capture	Ref.
				pH	Q [m^3^/L·h]	T [°C]			
Industrial WW ^1^	~2000	Lime	Packed tower	11.0	420	25	99%	NR ^2^	[139]
Acetylene Purification WW	125	NaOH	Stripping column	12.0	0.5	60	91%	H_2_SO_4_ 0.2 mol/L	[140]
Landfill Leachate	1158	Lime, NaOH,	Raschig ring-packed column	12.0	0.9	25	98%	H_3_PO_4_ 0.24 mol/L	[141]
	2738	NaOH, Ca(OH)_2_	Packed tower	8.0–11	0.18–0.63	25–30	91–95.5%	H_2_SO_4_	[142]
Swine WW	~2000	Lime	Stripping tower	11.2	0.065	20	90%	NR	[143]
	4950	NaOH	Laboratory reactor	10.0	0.4	37	88.2%	50% H_2_SO_4_	[144]
Mixed Livestock WW	2.970	NaOH	Pilot stripping reactor	9.0–10.7	NR	30–40	82%	H_2_SO_4_	[145]
Anaerobic Digestate	7170	NaOH, Ca(OH)_2_	Stripping column	10.0	0.2–0.5	40–70	91–96%	H_2_SO_4_	[138]
	297.6	MgO	Stripping column	10.0	0.48	40	94.34%	H_3_PO_4_	[146]
Human Urine	1963.5	NaOH	Laboratory reactor	11–13.5	0.09–0.27	16	98.7%	H_2_SO_4_ 0.5 mol/L	[147]
	~4000	NaOH	Stripping column	9.3–11	0.12–0.84	25–60	97%	H_2_SO_4_ 1 mol/L	[148]

^1^—wastewater; ^2^—not reported. Innovative approaches to ammonia air stripping include hybrid systems that combine various technologies, such as electrodialysis (ED) with electrochemical ammonia stripping (EAS). An example is the study [149], which employed an ED-EAS system for selective ammonia recovery from post-digestion digestate. This technology concentrated ammonium ions to 3.775 g/L, a threefold increase compared to the initial concentration. The process achieved 90.5% ammonia recovery with an energy consumption of 11.6 kWh/kg-NH_3_ during the EAS phase. Integrating ED and EAS processes enhances recovery efficiency, reduces operational costs, and enables the production of high-value products such as ammonium sulfate, aligning with the principles of a circular economy. Another type of hybrid system utilizes waste heat from biogas systems (CHP), significantly lowering the energy costs of ammonia stripping [145]. Modular stripping reactors tailored to various wastewater sources facilitate efficient ammonia removal, achieving energy consumption levels of 0.52 kWh/kg-N and producing mineral fertilizers that meet EU standards of Regulation (EU) 1009/2019 [145]. Current research focuses on integrating ammonia recovery processes with renewable energy sources, further optimizing operational efficiency, and utilizing by-products to improve economic performance.

**Table 4 materials-18-01422-t004:** Parameters and efficiency of steam stripping for ammonia removal from different types of wastewater.

Wastewater Type	Ammonia Concentr. [mg/L]	Reactor Design	Operational Conditions			$\eta_{{NH}_{3}}$	Ref.
			G/L	T [°C]	Other		
Metallurgical Industry Wastewater	~5200	Small-scale pilot countercurrent steam stripping reactor	~2.55 lb/gal	~101	pH: 11.5	up to 99.9%	[153]
Process Water with Nitrobenzene contamination	Not specified	Oldershaw glass column, 10–20 sieve trays, φ = 1 inch	~0.065 kg/kg	~100	NR ^1^	up to 99.8%	[151]
Shale Oil Wastewater	1065–24,689	Packed bed column with countercurrent flow	~1.0 kg/kg	~93.3	NR	66.5–99%	[154]
Cattle Manure Digestate	910–45,000	Ceramic-filled steam stripping tower	~63 kg/m^3^ ~11,000 mL/min	~98.5	pH: ~9.6	91–96%	[155]
Separately Collected Human Urine	4200–9200	Pall ring-packed tower (surface area: 360 m^2^/m^3^)	~25 kg/h	~100	Urine flow: ~90 L/h; pH: ~9.25	83–96.8%	[156]
Digestate and Centrate from Anaerobic Digestion	950–4700	Column with perforated trays, 11 levels, vacuum system	~4.5	~81	Vacuum: ~0.75 m H_2_O	90–98%	[157]
Ammonia-Rich Wastewater	5000–20,000	Rotating packed bed (RPB); compact	~0.18 kg/kg	~135 (steam), ~55 (liquid)	RPM: ~750; pH > 11	95–98.8%	[150]

^1^—Not reported.

**Table 5 materials-18-01422-t005:** Comparison of conventional mesophilic biological nitrogen removal (BNR) processes and thermophilic-membrane systems.

Parameter	Thermophilic System with Membranes	Conventional Mesophilic BNR
Operating Temperature	50–65 °C	20–37 °C
Microbial Activity	Dominated by thermophilic bacteria	Dominated by mesophilic bacteria
Ammonia Volatilization	Enhanced at high temperatures	Minimal
Nitrification Efficiency	Partial or limited due to thermal inhibition	High under optimal conditions
Sludge Production	Reduced due to enhanced biodegradation	Higher sludge yield
Membrane Integration Benefits	Biomass retention, nitrogen concentration	Less commonly used
Energy Demand	Higher due to heating	Lower but requires aeration
Pathogen Removal	High due to thermophilic conditions	Moderate

**Table 6 materials-18-01422-t006:** Parameters and efficiency of electrochemical reduction in nitrates from different types of wastewater.

Wastewater Type	Reactor Design	Process Conditions	Additives	$\eta_{{NH}_{3}}$	Ref.
Wastewater containing nitrates; 0.05 M NaNO_3_	Teflon chamber divided by Nafion 117 cation-exchange membrane; Bi cathode (2 cm^2^), Pt anode (6 cm^2^)	Potential: from −1.8 to −2.6 V vs. Ag/AgCl; constant helium flow (12 mL/min)	0.4 M NaHCO_3_; 0.4 M Na_2_CO_3_	Selectivity to N_2_: 58–65%; total Faradaic efficiency: 18–79%	[200]
Synthetic nitrate solution; 100 mg/L NO_3_^−^	Undivided cylindrical acrylic reactor, 400 mL volume; Cu–Zn cathode (62.2% Cu, 37.8% Zn), Ti/IrO_2_–Pt anode	Current: 40 mA/cm^2^; initial pH: 3.0, 6.5 or 12.0; duration: 300 min	0.50 g/L NaCl; 0.5 g/L Na_2_SO_4_,	Removal of NO_3_^−^–N from 100 mg/L to 9.7 mg/L, no NH_3_ or NO_2_^−^ detected	[201]
Synthetic nitrate solution; 2–10 mM NaNO_3_	Rotating disk electrode with single-crystal copper; Cu (100) and Cu (111) single-crystal electrodes	Current: dependent on potential (+0.15 to −0.5 V vs. RHE), 50 mV/s scanning	0.1 M HClO_4_; 0.1 M NaOH	Cu (100): higher activity for reduction to NH_2_OH in alkaline pH	[202]
Industrial wastewater; 200 mg/L NO_3_^−^	Reactor divided by Nafion 117 membrane; cathode: carbon with biofilm, anode: graphite	Voltage: −0.6 V; temperature: 30 °C; duration: 5 h	Na_2_CO_3_ buffer	96% nitrate removal, minimal NH_3_ production	[203]
Industrial wastewater with high nitrate concentration; up to 85,000 mg/L NO_3_^−^	Various technologies: ER, EC, ED; electrodes: Cu, Sn, Fe, Al, BDD	Voltages: from −1.1 V to 15 V; pH: from 3 to 12	NaCl; SO_4_^2−^; NaOH	Efficiency up to 100% depending on technology	[204]
Agro-industrial and synthetic wastewater; 27 g/L NO_3_^−^	Two-chamber reactor with cation-exchange membrane; BDD cathodes and anodes	Current: 300–750 A/m^2^; temperature: 20–34 °C; low pH	pH = 1.5	Nitrate reduced to <50 mg/L, no NH_3_ or NO_2_^−^	[205]

**Table 7 materials-18-01422-t007:** Parameters and efficiency of electrochemical oxidation of ammonia from different types of wastewater.

Wastewater Type	Reactor Design	Process Conditions	Additives	$\eta_{{NH}_{3}}$	Ref.
Dyeing wastewater; 161.2 ± 2.0 mg-N/L	Electrochemical cell, 0.4 L; anode: Ti/PbO_2_, cathode: Ti, electrode gap 20 mm	Current density: 20 mA/cm^2^; flow: 300 mL/min; pH: 8.3; temperature: 30 °C	1.0 g/L NaCl	100% in 60 min; N_2_ selectivity: 88.3%	[209]
Coking wastewater; NH_3_-N: 10 mg/L; TOC: 78 mg/L	Batch cell, 250 mL, controlled temperature; anode: Ti/BDD; cathode: stainless steel, electrode gap 10 mm	Current density: 20–60 mA/cm^2^; temperature: 30–60 °C	0.2 M Na_2_SO_4_	Complete removal under neutral and alkaline conditions	[210]
Simulated wastewater; NH_3_-N: 2000 mg/L	Non-diaphragm electrolysis cell, 0.5 L; anode and cathode: graphite, electrode gap 40 mm	Current density: 90 mA/cm^2^; pH 10; time: 8 h	8000 mg/L NaCl	Reduction from 2000 to 280 mg/L (86%)	[212]
Synthetic and anaerobic digester effluent; NH_3_-N: 900 mg/L (synthetic); 560 mg/L (effluent)	Electrochemical cell, 570 mL, airflow 20 L/min; anode: Ti/RuO_2_-IrO_2_, cathode: porous titanium, floating cathode configuration	Current density: 5 mA/cm^2^; pH: 6.77; time: 8 h	0.2 g/L NaCl	Up to 216 mg NH_3_-N/L/h (synthetic); 110 mg NH_3_-N/L/h (anaerobic effluent)	[213]
Municipal and synthetic wastewater; NH_3_-N: 40 mg/L (synthetic); 60 mg/L (municipal)	Acrylic cell, 300 mL; anode: Ti/RuO_2_-Pt; cathode: Ti, electrode gap 1 cm	Current density: 20 mA/cm^2^; time: 30 min	0.5 g/L NaCl	88.3% (municipal); 87.5% (simulated)	[211]

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
