# Peer review of "Advanced Technologies for Nitrogen Removal and Recovery from Municipal and Industrial Wastewater"

_materials, 2025, doi:10.3390/ma18071422_

Round 1
Reviewer 1 Report
Comments and Suggestions for Authors
Thermophilic systems coupled with ultrafiltration membranes can play a significant role in the nitrogen cycle, particularly in the transformation of organic nitrogen into ammoniacal nitrogen, which is easier to recover or remove using conventional methods.
A more in-depth analysis of the fate of nitrogen within biological sludge is required, as wastewater treatment can no longer be discussed without considering its residual by-products. Specifically, regarding the nitrogen cycle and resource recovery, sludge minimization techniques (such as combustion) lead to a significant loss of nitrogen into the atmosphere, making it less bioavailable.
A table summarizing the main factors influencing the nitrification and denitrification process should be included, as this would help in structuring the presented results.
Additionally, the introduction should include a reference to the presence of synthetic nitrogen fertilizers, which have profoundly altered the way nitrogen resources are valued, as nitrogen was historically recovered from wastewater.
In the section on environmental impact, it is necessary to quantify the greenhouse effect potential of nitrogen oxides, which often escape the treatment process as by-products.

Author Response
Reviewer 1:
Comment #1:
Thermophilic systems coupled with ultrafiltration membranes can play a significant role in the nitrogen cycle, particularly in the transformation of organic nitrogen into ammoniacal nitrogen, which is easier to recover or remove using conventional methods.
Respond:
We appreciate the reviewer's insightful comment regarding the potential of thermophilic systems coupled with ultrafiltration membranes to enhance nitrogen cycling, specifically the transformation of organic nitrogen to ammoniacal nitrogen. We concur that this approach holds significant promise for optimizing nitrogen management in wastewater treatment. Therefore, an additional paragraph has been added to section 3.3, the Membrane Technologies chapter.
“One of the most perspective approaches is linking of thermophilic systems with membranes. These systems combines the advantages of high-temperature biological processes and membrane separation technology to enhance treatment efficiency and resource recovery. Compared to conventional mesophilic biological nitrogen removal (BNR) processes, the thermophilic-ultrafiltration system presents distinct advantages and challenges (Table 5). By concentrating nitrogen-rich compounds in the retained sludge, the system facilitates downstream recovery techniques such as struvite precipitation or ammonia strip-ping. The membrane application provides a physical barrier to suspended solids and pathogenic microorganisms, improving effluent quality and making the treated water suitable for reuse applications. Moreover the membrane effectively retains thermophilic microbial populations within the bioreactor, promoting higher sludge retention times (SRT) and increasing process stability [208].
While thermophilic systems require additional energy input for heating, this can be offset by the potential for energy recovery (e.g., from anaerobic digestion) and the reduction in sludge disposal costs. Moreover, the integration with membrane processes significantly enhances process stability and effluent quality, making the system viable for applications in high-strength wastewater treatment (e.g., industrial effluents, landfill leachate, or live-stock manure [190].
Table 5. Comprehension of to conventional mesophilic biological nitrogen removal (BNR) processes and the thermophilic-membrane systems.
|
Parameter |
Thermophilic System with membranes |
Conventional Mesophilic BNR |
|
Operating Temperature |
50–65°C |
20–37°C |
|
Microbial Activity |
Dominated by thermophilic bacteria |
Dominated by mesophilic bacteria |
|
Ammonia Volatilization |
Enhanced at high temperatures |
Minimal |
|
Nitrification Efficiency |
Partial or limited due to thermal inhibition |
High under optimal conditions |
|
Sludge Production |
Reduced due to enhanced biodegradation |
Higher sludge yield |
|
Membrane Integration Benefits |
Biomass retention, nitrogen concentration |
Less commonly used |
|
Energy Demand |
Higher due to heating |
Lower but requires aeration |
|
Pathogen Removal |
High due to thermophilic conditions |
Moderate |
“
Comment #2:
A more in-depth analysis of the fate of nitrogen within biological sludge is required, as wastewater treatment can no longer be discussed without considering its residual by-products. Specifically, regarding the nitrogen cycle and resource recovery, sludge minimization techniques (such as combustion) lead to a significant loss of nitrogen into the atmosphere, making it less bioavailable.
Respond:
Thank you for your insightful comment. We have expanded our manuscript of the fate of nitrogen within biological sludge, incorporating a more detailed discussion on its role in the nitrogen cycle and resource recovery. Specifically, we now address how different sludge treatment and minimization techniques, such as combustion and anaerobic digestion influence nitrogen retention and loss. In particular, we highlight the atmospheric nitrogen losses from combustion processes and their implications for bioavailability. This new chapter of the manuscript strengthens the discussion on wastewater treatment sustainability and aligns with modern resource recovery perspectives. We appreciate your valuable feedback.
The following new chapter has been added.
“3.6. Fate of nitrogen within biological sludge treatment”
Comment #3:
A table summarizing the main factors influencing the nitrification and denitrification process should be included, as this would help in structuring the presented results.
Respond:
Thank you for your suggestion. We have now included a Table 2 summarizing the main factors influencing the nitrification and denitrification processes. This table provides an overview of key parameters such as temperature, pH, dissolved oxygen levels along with their effects on process efficiency. We believe this addition helps structure the presented results and improves clarity. We appreciate your feedback in enhancing the comprehensibility of our study.
Comment #4:
Additionally, the introduction should include a reference to the presence of synthetic nitrogen fertilizers, which have profoundly altered the way nitrogen resources are valued, as nitrogen was historically recovered from wastewater.
Respond:
Thank you for the insightful comment. We have incorporated this suggestion into the introduction to better reflect the historical shift in nitrogen resource management due to the introduction of synthetic nitrogen fertilizers. These changes have been implemented in lines 40–42 of the revised manuscript.
Comment #5:
In the section on environmental impact, it is necessary to quantify the greenhouse effect potential of nitrogen oxides, which often escape the treatment process as by-products.
Respond:
Thank you for your valuable feedback. We have now included a quantification of the greenhouse effect potential of nitrogen oxides (NOₓ) in the section on environmental impact. Specifically, we have added information on the global warming potential (GWP) of NOₓ and its contribution to radiative forcing. Additionally, we have discussed the typical emission levels from the treatment process and their environmental implications. We appreciate your suggestion, as it helps strengthen our discussion on the environmental impact of our study.
The following description was added at the end of 1.3. Environmental impact and toxicity of various nitrogen forms line [477-485]:
Despite the negative impact of nitrogen forms on aquatic ecosystems it is necessary to quantify the greenhouse effect potential of nitrogen oxides, which often escape the treat-ment process as by-products. Specifically, nitrous oxide (N₂O), a significant nitrogen oxide emitted during wastewater treatment, has a global warming potential (GWP) approximately 265 times that of carbon dioxide (CO₂) over a 100-year timeframe. This means that even small emissions of N₂O can have a substantial impact on climate change. [84]. While other nitrogen oxides like NO and NO₂ also contribute to atmospheric chemistry, their direct GWP is less significant compared to N₂O. However, they play a crucial role in the formation of ground-level ozone, another greenhouse gas, and contribute to acid rain.
Reviewer 2 Report
Comments and Suggestions for Authors
Overall, the review at present version is highly descriptive and contains several fundamental information which are already known to the readers. The whole draft should be carefully revised and condensed to make the review more concise, meaningful and impactful. Addition of more tables on the important data on nitrogen removal performance of different technologies as well as colorful figures would improve the quality of the review.
Comments:
Title: It is not accurately conveying the main scope of this review. Are you trying to say “Advanced Technologies for Nitrogen Removal and ….”
In fourth paragraph, report the typical concentrations range of different nitrogen species found in municipal wastewater.
Line 95 – 96: “Under natural conditions, nitrites (NO₂⁻) and nitrates (NO₃⁻) are formed through a two-step nitrification process…”. For better understanding, add the associated biochemical reactions.
Introduction: Authors are suggested to abridge the texts by deleting the less important texts to make the introduction more concise and readable.
Line 127: “1.4. Objectives and scope of the review”. In this section, highlight the novelty and importance of this review with respect to the existing reviews in literature. Justify what the review should be considered for publication.
Add a section on the “Review Method” by describing how literature analysis was done as well as what screening criteria were considered for collection of relevant references to prepare this review, e.g., the types of scientific database chosen, the keywords used in the database search engine, the target publications years (last 5, 10 or 15 years), etc.
Line 522: “Impact of environmental factors on the activity of nitrifying microorganisms”. Make appropriate sub-heading to list different types of factors to make visible to the readers. Also, prepare a table by summarizing the recent findings on influence of different factors on the nitrifying performance.
Tables 1 – 3: It looks incomplete at the current version, thus needs to be approbatively adjusted.
You have reported different nitrogen removal technologies. A section on the comparative discussion of advantages and limitations of each method would be exciting to the readers.
Author Response
Reviewer 2.
General Comment :
Overall, the review at present version is highly descriptive and contains several fundamental information which are already known to the readers. The whole draft should be carefully revised and condensed to make the review more concise, meaningful and impactful. Addition of more tables on the important data on nitrogen removal performance of different technologies as well as colorful figures would improve the quality of the review.
Respond:
We appreciate the reviewer's feedback regarding the descriptive nature and length of the current manuscript. We acknowledge the concern that certain sections contain fundamental information already known to the readers, and we agree that a more concise and impactful review is desirable.
As suggested, we incorporated comprehensive tables summarizing important data on the nitrogen removal performance of various technologies. These tables provide a clear and concise overview of the effectiveness and limitations of different approaches. We shifted the focus from descriptive reporting to critical analysis, highlighting the knowledge gaps, challenges, and future directions in nitrogen management. We believe that these revisions significantly improved the conciseness, meaningfulness, and impact of the review.
Comment #1:
Title: It is not accurately conveying the main scope of this review. Are you trying to say “Advanced Technologies for Nitrogen Removal and ….”
Respond:
Thank you for this suggestion. After carefully reviewing the scope of our study and considering your feedback regarding the clarity of the title, we have decided to revise it to better reflect the content of our review.
The new title is: “Advanced Technologies for Nitrogen Removal and Recovery from Municipal and Industrial Wastewater”. This revised title highlights both nitrogen removal and recovery, which are the core topics of our review.
Comment #2:
In fourth paragraph, report the typical concentrations range of different nitrogen species found in municipal wastewater.
Respond:
We thank the reviewer for pointing out the need for specific concentration ranges of nitrogen species in municipal wastewater. In response, we added Table 1 with required information.
Table 1. Typical concentrations of the nitrogen species in the municipal [213].
|
Nitrogen Species |
Typical Concentration Range (mg/L) |
Remarks |
|
Total Nitrogen (TN) |
20 – 85 |
Sum of all nitrogen species, including organic and inorganic forms. |
|
Organic Nitrogen |
8 – 35 |
Includes proteins, amino acids, urea and humic substances. |
|
Ammonium (NH4+) |
12 – 50 |
Predominant form of nitrogen in wastewater due to urea hydrolysis. |
|
Nitrite (NO2-) |
<0.1 – 0.5 |
Intermediate oxidation product, usually present in very low concentrations. |
|
Nitrate (NO3−) |
<0.1 – 1.0 |
Minimal in raw wastewater, but increases after nitrification in treatment processes. |
|
Dissolved Organic Nitrogen (DON) |
1 – 10 |
Comprises low-molecular-weight organic nitrogen compounds. |
Comment #3:
Line 95 – 96: “Under natural conditions, nitrites (NO₂⁻) and nitrates (NO₃⁻) are formed through a two-step nitrification process…”. For better understanding, add the associated biochemical reactions.
Respond:
Thank you for your valuable suggestion. We agree that adding biochemical reactions can improve clarity. These reactions are already presented and discussed in detail in Section 2.1. However, to enhance readability, we have revised the relevant paragraph to ensure a clearer presentation of the nitrification process.
Comment #4:
Introduction: Authors are suggested to abridge the texts by deleting the less important texts to make the introduction more concise and readable.
Respond:
Thank you for your valuable suggestion. In response to your comment, we have carefully reviewed the introduction and removed less essential information to improve conciseness and readability. The revised version retains the necessary context while making the section more focused and engaging for readers.
Comment #5:
Line 127: “1.4. Objectives and scope of the review”. In this section, highlight the novelty and importance of this review with respect to the existing reviews in literature. Justify what the review should be considered for publication.
Respond:
Thank you for your valuable feedback. We have revised Section 1.4, "Objectives and Scope of the Review," to explicitly highlight the novelty and significance of our review compared to existing literature. In particular, we have:
Emphasized Novelty: We have clarified how our review differs from and advances beyond previous reviews by incorporating recent developments, addressing specific gaps, and synthesizing emerging trends. Highlighted Importance: We have elaborated on the critical relevance of our review to the field, demonstrating how it provides a comprehensive, up-to-date perspective that benefits both researchers and practitioners. Justified Publication: We have explained why our review offers a unique contribution that warrants publication, including its potential to shape future research directions and practical applications.
Comment #6:
Add a section on the “Review Method” by describing how literature analysis was done as well as what screening criteria were considered for collection of relevant references to prepare this review, e.g., the types of scientific database chosen, the keywords used in the database search engine, the target publications years (last 5, 10 or 15 years), etc.
Respond:
Thank you for your valuable remark. We have addressed this comment by adding a clear description of the review methodology directly into section 1.4 “Objectives and scope of the review” (page 9). In this section, we clearly indicate the scientific databases used (Scopus, Web of Science, ScienceDirect, Google Scholar), publication selection criteria, and the target publication years (priority given to literature from the past 10 years).
Comment #7:
Line 522: “Impact of environmental factors on the activity of nitrifying microorganisms”. Make appropriate sub-heading to list different types of factors to make visible to the readers. Also, prepare a table by summarizing the recent findings on influence of different factors on the nitrifying performance.
Respond:
Thank you for your suggestion. We have now included a Table 2 summarizing the main factors influencing the nitrification and denitrification processes. This table provides an overview of key parameters such as temperature, pH, dissolved oxygen levels along with their effects on process efficiency. We believe this addition helps structure the presented results and improves clarity. We appreciate your feedback in enhancing the comprehensibility of our study.
Comment #8:
Tables 1 – 3: It looks incomplete at the current version, thus needs to be approbatively adjusted.
Respond:
Indeed, during the manuscript generation process following the template, an automatic removal of section dividers occurred, leading to unintended errors in the display of the full content of Tables 1–4. This issue was beyond our control but has now been corrected. We sincerely apologize for this problem.
Comment #9:
You have reported different nitrogen removal technologies. A section on the comparative discussion of advantages and limitations of each method would be exciting to the readers.
Respond:
We appreciate the reviewer's suggestion to include a comparative discussion of the advantages and limitations of the various nitrogen removal technologies presented. We agree that this would significantly enhance the review's practical value and provide readers with a more comprehensive understanding of the trade-offs associated with each method. In response, we add a dedicated paragraph entitled: 3.5 Technology readiness and scalability of nitrogen removal and recovery methods.
Reviewer 3 Report
Comments and Suggestions for Authors
The authors present a review where is supposed to study different Nitrogen removal methods, but they remain a general study without going into the work done on each of the techniques.
Several Polish words are inside the text which make difficult to follow some parts (see Table 2 caption)
Some tables seem to be uncomplete and the data included are not supported by references. In tables 1 and 2 footnotes are together with the rest of the text.
In my opinion, a review should provide the latest research to date, compiling the most current work in each of the sections. This has not been done in this review. For example, the section on membranes talks about generalities of membrane processes, none of the references allude to work done to retain or remove NH4/NH3. And something similar in the other methods discussed.
Therefore, I believe that this review should not be published in that version. The authors should redo it and reorient it completely
Author Response
Reviewer 3:
Comment #1:
The authors present a review where is supposed to study different Nitrogen removal methods, but they remain a general study without going into the work done on each of the techniques.
Answer:
Thank you for your comment. We acknowledge the importance of providing a more detailed discussion of specific studies related to each nitrogen removal technique. In our initial submission, much of this information was included in Tables 1–5, but due to formatting issues, key references and data were not clearly visible.
To address this, we have reformatted and refined the tables, ensuring that they now clearly present the relevant studies. Additionally, we have expanded the discussion in the main text by incorporating specific research findings, quantitative performance data, and case studies for each method. These improvements enhance the depth of analysis and provide a more comprehensive review of nitrogen removal technologies.
Comment #2:
Several Polish words are inside the text which make difficult to follow some parts (see Table 2 caption).
Answer:
Thank you for pointing this out. We have carefully reviewed the manuscript and corrected all occurrences of Polish words, including the caption under Table 2, to ensure consistency and readability in English.
Comment #3:
Some tables seem to be uncomplete and the data included are not supported by references. In tables 1 and 2 footnotes are together with the rest of the text.
Answer:
During the manuscript generation process following the template, an automatic removal of section dividers occurred, leading to unintended errors in the display of the full content of Tables 1–4. This issue was beyond our control but has now been corrected. We sincerely apologize for this problem.
Comment #4:
In my opinion, a review should provide the latest research to date, compiling the most current work in each of the sections. This has not been done in this review. For example, the section on membranes talks about generalities of membrane processes, none of the references allude to work done to retain or remove NH4/NH3. And something similar in the other methods discussed.
Answer:
Thank you for your comment. We recognize the importance of including recent studies to ensure the relevance of our review. In our original submission, Tables 1–4 contained 29 citations, primarily referencing research from the last few years. However, due to formatting issues, these references were not clearly visible. To address this, we have corrected and reformatted the tables, ensuring that all relevant studies are properly displayed. Additionally, we have reviewed and updated key sections to further emphasize recent advancements in nitrogen removal technologies. We appreciate your careful review and your constructive feedback.
Reviewer 4 Report
Comments and Suggestions for Authors
This work shares a review work on availability of nitrogen and the methods for its recovery from wastewaters fundamentally by (de)nitrification and physical and chemical processes. A comprehensive overview on present status of the technologies and underlying chemical or physical mechanisms taking place are discussed, including soe emerging technologies based on the process intensification approach. The paper brings coordinated discussions flowing well between the first chapters on the background for availability for nitrogen (first nine pages) and chemical processing for removal in the second part. The paper is well presented, and the chapters are well written and therefore is worth publication. which remain at R&D stage.
Some few comments could also be completed by the authors.
- In the first section on nitrogen availability and environmental impact , it would have been good to add a section on seasonal variations and regulatory limits in water and soil and how recent changes in human activities and global waring is affecting the contents and consequences on regulatory amendments on policy limits.
- Since nitrification by microorganisms and process design is strongly influenced by the process kinetics which impacts reactor design, it would good if the authors report more details on the modelling of the kinetics nitrates to nitrites which impact reactor sizing
- Nn need to specify in page 15, line 750 the air flow of 0.12-0.9 m3/L since this value is function of the size of the process and the scale of the operations
- Page 17, Table 2. Please the title (to be written in English). Also, the size of Table 2 does not fit the page format.
- Page 17. Woud it be possible to add models on process maturity of reported techniques and scaleup?
- Page 21. It is not clear how relevant are the models (9-11) of energy for membrane while they not mentioned for other processes (classical (ex. stripping) or advanced ones (UV/microwave etc.. assisted) other processes.
- Page 21 , line 102. The word “gdzie” to arrange.
- Tables 3 and 4 the size needs to be adapted to the page
- A paragraph on a comparative study between the processes biological and chemical would be good to add as summary of the work where further scaleup and economics could be added.
The level of English is fine. but some typos need to be adressed.
Author Response
Reviewer 4.
General comment
This work shares a review work on availability of nitrogen and the methods for its recovery from wastewaters fundamentally by (de)nitrification and physical and chemical processes. A comprehensive overview on present status of the technologies and underlying chemical or physical mechanisms taking place are discussed, including soe emerging technologies based on the process intensification approach. The paper brings coordinated discussions flowing well between the first chapters on the background for availability for nitrogen (first nine pages) and chemical processing for removal in the second part. The paper is well presented, and the chapters are well written and therefore is worth publication which remain at R&D stage.
Respond:
Thank you for your positive assessment and valuable feedback. We appreciate your recognition of the manuscript’s structure and content. Your comments encourage us, and we acknowledge that some aspects remain at the research and development stage.
Comment #1:
In the first section on nitrogen availability and environmental impact , it would have been good to add a section on seasonal variations and regulatory limits in water and soil and how recent changes in human activities and global waring is affecting the contents and consequences on regulatory amendments on policy limits.
Thank you for your insightful suggestion. We recognize the importance of addressing seasonal variations in nitrogen availability, regulatory limits in water and soil, and the impact of recent changes in human activities and global warming. In response to your feedback, we have made the following additions to the first section:
- Seasonal Variations: We have included a discussion on how nitrogen availability fluctuates across different seasons due to variations in temperature, precipitation, and biological activity. This provides a more comprehensive understanding of nitrogen dynamics.
- Regulatory Limits and Recent Changes: We have expanded the section to include an overview of existing regulatory limits for nitrogen in water and soil, how they vary by region, and recent amendments in response to environmental concerns.
- Impact of Human Activities and Global Warming: We have incorporated recent findings on how climate change and anthropogenic influences—such as intensified agriculture, urbanization, and industrial emissions—are altering nitrogen levels, leading to shifts in policy regulations.
Comment #2:
Since nitrification by microorganisms and process design is strongly influenced by the process kinetics which impacts reactor design, it would good if the authors report more details on the modelling of the kinetics nitrates to nitrites which impact reactor sizing
Respond:
We appreciate the reviewer's comment regarding the importance of nitrification kinetics in reactor design, particularly the conversion of nitrite to nitrate. We agree that providing more detail on the modeling of these kinetics is crucial for a comprehensive understanding of reactor sizing and process optimization. The dedicated section “Kinetic Modeling of Nitrification and Its Impact on Reactor Sizing” has been added from the line 664 in new version of the manuscript.
Comment #3:
Nn need to specify in page 15, line 750 the air flow of 0.12-0.9 m3/L since this value is function of the size of the process and the scale of the operations
Respond:
Thank you for your insightful comment. We acknowledge that the airflow rate (0.12–0.9 m³/(h·L)) is dependent on the size of the system and operational scale. To provide better clarity, we have revised the sentence accordingly to reflect this variability.
Comment #4:
Page 17, Table 2. Please the title (to be written in English). Also, the size of Table 2 does not fit the page format.
Respond:
Thank you for your valuable feedback. The title of Table 2 (now Table 4) has been translated into English, and the table format has been adjusted to ensure proper alignment with the page layout.
Comment #5:
Page 17. Woud it be possible to add models on process maturity of reported techniques and scaleup?
Respond:
Thank you for your valuable suggestion. We agree that incorporating models on the process maturity of reported techniques and their scale-up potential would enhance the discussion. In response, we have added whole new section entitled 3.5 Technology readiness and scalability of nitrogen removal and recovery methods.
Comment #6:
Page 21. It is not clear how relevant are the models (9-11) of energy for membrane while they not mentioned for other processes (classical (ex. stripping) or advanced ones (UV/microwave etc.. assisted) other processes.
Respond:
Thank you for this valuable remark. The energy consumption models (equations 9–11) were specifically included for membrane-based processes due to their critical role in operational cost estimation, optimization, and scale-up considerations, as these are commonly reported and standardized in membrane technology literature. In contrast, for classical processes (e.g., ammonia stripping) or advanced technologies (UV/microwave-assisted methods), energy consumption data are usually reported directly from empirical studies rather than derived from universal or standardized predictive equations.
Comment #7:
Page 21 , line 102. The word “gdzie” to arrange.
Respond:
Thank you for pointing this out. The error in line 102 has been corrected to ensure proper wording and clarity.
Comment #8:
Tables 3 and 4 the size needs to be adapted to the page.
Respond:
This issue was beyond our control but has now been corrected. We sincerely apologize for this problem.
Comment #9:
A paragraph on a comparative study between the processes biological and chemical would be good to add as summary of the work where further scaleup and economics could be added.
Respond:
Thank you for your insightful suggestion. We recognize the importance of a comparative analysis between biological and chemical processes, particularly in terms of scalability and economic feasibility. Such issues have been addressed in the new section “3.5 Technology readiness and scalability of nitrogen removal and recovery methods”.
Round 2
Reviewer 2 Report
Comments and Suggestions for Authors
The submitted revision looks accurate.
Author Response
Comments 1: The submitted revision looks accurate.
Response 1: Thank you for your positive assessment. We appreciate your time and valuable feedback throughout the review process.
Reviewer 3 Report
Comments and Suggestions for Authors
The article has clearly been improved and the study is now more comprehensive, which should have been done in the first version.
There are now some editing errors that need to be corrected before publication:
- Page 20, 25 … are indexed as page 2 of 52
- Page 25: equation (8) should be eq (9) and the following equations are to be renumbered
- There are several references that are uncompleted as 184, 185 …
Author Response
Comments 1: The article has clearly been improved and the study is now more comprehensive, which should have been done in the first version.
Response 1: Thank you for your feedback. We appreciate your recognition of the improvements made to the manuscript.
Comments 2: There are now some editing errors that need to be corrected before publication: Page 20, 25 … are indexed as page 2 of 52; Page 25: equation (8) should be eq (9) and the following equations are to be renumbered; There are several references that are uncompleted as 184, 185 …
Response 2: We have carefully reviewed the manuscript and corrected the identified editing errors:
- The page numbering issue on pages 20 and 25 has been fixed.
- Equation (8) has been renumbered as Equation (9), and the subsequent equations have been adjusted accordingly.
- Incomplete references have been completed and verified for accuracy.